# Sparse Weight Averaging with Multiple Particles for Iterative Magnitude Pruning

**Moonseok Choi**[*1]  **Hyungi Lee**[*1]  **Giung Nam**[*1]  **Juho Lee**[1,2]
[1]KAIST AI  [2]AITRICS
{ms.choi, lhk2708, giung, juholee}@kaist.ac.kr

## Abstract

Given the ever-increasing size of modern neural networks, the significance of sparse architectures has surged due to their accelerated inference speeds and minimal memory demands. When it comes to global pruning techniques, Iterative Magnitude Pruning (IMP) still stands as a state-of-the-art algorithm despite its simple nature, particularly in extremely sparse regimes. In light of the recent finding that the two successive matching IMP solutions are linearly connected without a loss barrier, we propose Sparse Weight Averaging with Multiple Particles (SWAMP), a straightforward modification of IMP that achieves performance comparable to an ensemble of two IMP solutions. For every iteration, we concurrently train multiple sparse models, referred to as particles, using different batch orders yet the same matching ticket, and then weight average such models to produce a single mask. We demonstrate that our method consistently outperforms existing baselines across different sparsities through extensive experiments on various data and neural network structures.

## 1 Introduction

Deep neural networks are often highly over-parameterized, and the majority of their parameters can be pruned without sacrificing model performance (LeCun et al., 1989). The *lottery ticket hypothesis* proposed by Frankle & Carbin (2019) suggests that there exists a sparse subnetwork at initialization that can be trained to achieve the same level of performance as the original dense network. Such *matching* subnetworks can be found via Iterative Magnitude Pruning (IMP) with rewinding (Frankle et al., 2020), which involves the following three steps: (i) training the network for a certain number of iterations, (ii) pruning weights with the smallest magnitudes, and (iii) rewinding the weights back to an early iteration while fixing the pruned weights to zero. This procedure is repeated for several rounds, and the final rewound subnetwork corresponds to the *matching ticket* that can achieve the performance of the full network. Despite its simplicity, IMP offers the state-of-the-art performance as to finding a sparse mask, especially for extreme sparsity regimes (Renda et al., 2020).

The success of IMP is indeed counter-intuitive considering its simplicity. In this regard, Frankle et al. (2020) revealed an underlying connection between the lottery ticket hypothesis and linear mode connectivity, indicating that the effectiveness of IMP is reliant upon its stability to stochastic optimization; IMP solutions reside in the same basin of attraction in the loss landscape. Delving deeper into the subject matter, Paul et al. (2023) found that linear mode connectivity also exists between successive IMP solutions with different sparsity levels. More precisely, they concluded that IMP fails to find a *matching* subnetwork if the solutions from consecutive rounds are disconnected and further highlighted the significance of both the pruning ratio and the rewinding iteration to retain the connectivity between IMP solutions.

Inspired by the connection between IMP and linear mode connectivity, we expand the understanding to the *loss landscape perspective*. Analyzing the loss landscape of deep neural networks is an effective tool that is widely employed to study mode connectivity (Draxler et al., 2018; Garipov et al., 2018; Fort & Jastrzebski, 2019; Benton et al., 2021), and it also motivates us to find solutions located

---

* Equal contribution

at the flat region of the loss landscape to enhance generalization (Chaudhari et al., 2017; Izmailov et al., 2018; Foret et al., 2021). Notably, both fields share a common objective of identifying "good" subspaces characterized by low loss value, and this objective aligns with the ultimate goal of neural network pruning - to identify "matching" sparse subspaces within a given dense parameter space.

In this paper, we study how IMP can benefit from the multiple models connected in loss surfaces. Our contributions are summarized as follows:

- We first empirically demonstrate that multiple models trained with different SGD noise yet from the same matching ticket can be weight-averaged, i.e., there exists no loss barrier within the convex hull of the model weights. We further show that taking an average of the particles leads to flat minima, which exhibit superior generalization performance compared to each individual particle.

- Building upon prior observations, we propose a novel iterative pruning technique, Sparse Weight Averaging with Multiple Particles (SWAMP), tailored specifically for IMP. We verify that SWAMP preserves the linear connectivity of successive solutions, which is a crucial feature contributing to the success of IMP.

- Through extensive experiments, we provide empirical evidence that supports the superiority of the proposed SWAMP algorithm over other baselines.

## 2 BACKGROUNDS

### 2.1 NEURAL NETWORK PRUNING AS CONSTRAINED OPTIMIZATION

Conventional training of neural networks aims to find an optimal neural network parameter $\boldsymbol{w} \in \mathbb{R}^D$ that minimizes a given loss function $\mathcal{L} : \mathbb{R}^D \to \mathbb{R}$ for a given training dataset $\mathcal{D}$. Such optimization typically employs the Stochastic Gradient Descent (SGD; Robbins & Monro, 1951) methods, which we denote as $\boldsymbol{w}_T \leftarrow \mathrm{SGD}_{0 \to T}(\boldsymbol{w}_0, \xi, \mathcal{D})$ throughout the paper. Here, $\boldsymbol{w}_T$ denotes the solution obtained by performing SGD with a randomness of $\xi$ (e.g., mini-batch ordering) over $T$ iterations, starting from the initial weight $\boldsymbol{w}_0$. On the other hand, neural network pruning is the process of obtaining a sparse neural network with a desired sparsity level $\kappa \in [0, 1)$ from the original dense neural network. The goal is now to find an *optimal sparse solution* $\boldsymbol{w} = \boldsymbol{w}^* \circ \boldsymbol{m}^*$ subject to the constraint that the number of non-zero elements in the mask $\boldsymbol{m}^* \in [0, 1]^D$ satisfies $\|\boldsymbol{m}^*\|_0 \leq D(1 - \kappa)$.

### 2.2 ITERATIVE MAGNITUDE PRUNING WITH REWINDING

Iterative Magnitude Pruning (IMP; Frankle & Carbin, 2019) is an iterative pruning method that is both straightforward and highly effective. Each cycle of IMP involves the following three steps: *(i) Train* - a network parameter $\boldsymbol{w}_c$ at the $c^{\mathrm{th}}$ cycle is trained until it reaches convergence. *(ii) Prune* - a mask $\boldsymbol{m}$ is created by setting the smallest weights to zero based on a predefined pruning ratio of $\alpha$. *(iii) Reset* - the weights are then reverted back to their initial values before the next cycle begins. This *train-prune-reset* cycle is repeated until the desired level of sparsity is achieved.

However, in practical scenarios, the original version of IMP suffers from rapid performance degradation as sparsity increases and fails to match the performance of the original dense solution. To address this issue, the concept of *rewinding* is introduced (Frankle et al., 2020; Renda et al., 2020). Rather than *resetting* the unpruned weights to their initial values, the weights are *rewound* to an early training point - the *matching ticket*. The matching ticket is simply the weights obtained after training for a few iterations. Refer to Appendix C.1 for more details on IMP algorithms.

### 2.3 LINEAR CONNECTIVITY OF NEURAL NETWORK WEIGHTS

Consider a one-dimensional path denoted as $P : [0, 1] \to \mathbb{R}^D$, connecting two neural network weights $\boldsymbol{w}^{(0)}$ and $\boldsymbol{w}^{(1)}$ in a $D$-dimensional space, where the starting and end points are $P(0) = \boldsymbol{w}^{(0)}$ and $P(1) = \boldsymbol{w}^{(1)}$, respectively. In a simplified sense, we can say that there is a *connectivity* between $\boldsymbol{w}^{(0)}$ and $\boldsymbol{w}^{(1)}$ if the condition $\sup_{\lambda \in [0,1]} \mathcal{L}(P(\lambda)) \leq \max\{\mathcal{L}(P(0)), \mathcal{L}(P(1))\} + \epsilon$ holds, where $\epsilon$ is a small margin value. While recent advances in deep learning have revealed the existence of non-linear paths between local minima obtained through stochastic optimization (Draxler et al., 2018;

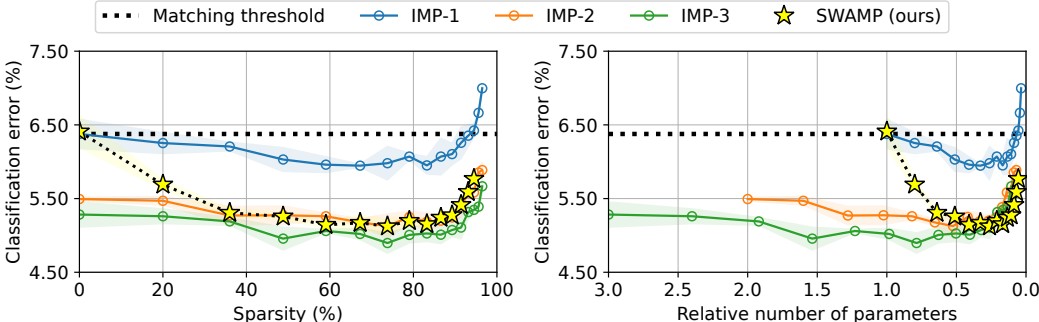

**Figure 1:** Classification error as a function of the sparsity (left) and the relative number of parameters (right). Our proposed SWAMP achieves remarkable performance comparable to an ensemble of IMP solutions, where IMP-$n$ indicates the ensemble of $n$ IMP solutions, while maintaining the same inference cost as a single model. Notably, SWAMP demonstrates *matching* performance even at extremely sparse levels, unlike IMP. The results are presented for WRN-28-2 on CIFAR-10, and we refer readers to Appendix C.6 for the same plot for CIFAR-100, as well as VGG architectures.

Garipov et al., 2018), it is still not straightforward to establish linear connectivity (i.e., connectivity with a linear connector $P(\lambda) = (1 - \lambda)\boldsymbol{w}^{(0)} + \lambda\boldsymbol{w}^{(1)}$) for modern deep neural networks (Lakshminarayanan et al., 2017; Fort & Jastrzebski, 2019; Fort et al., 2020).

## 3 SPARSE WEIGHT AVERAGING WITH MULTIPLE PARTICLES (SWAMP)

### 3.1 IMP: A LOSS LANDSCAPE PERSPECTIVE

Frankle et al. (2020) demonstrated that the matching ticket has a significant impact on the *stability* of neural networks to SGD noise $\xi$. Even when two networks are trained with the same random initialization $\boldsymbol{w}_0$, the different SGD noise $\xi^{(1)}$, $\xi^{(2)}$ disrupts the linear connectivity between the solutions obtained through SGD, i.e., there is no linear connector between

$$\boldsymbol{w}_T^{(1)} \leftarrow \mathrm{SGD}_{0 \rightarrow T}(\boldsymbol{w}_0, \xi^{(1)}, \mathcal{D}) \text{ and } \boldsymbol{w}_T^{(2)} \leftarrow \mathrm{SGD}_{0 \rightarrow T}(\boldsymbol{w}_0, \xi^{(2)}, \mathcal{D}), \tag{1}$$

and thus the optimization is rendered unstable to SGD noise. They further empirically confirmed that sparse solutions obtained through IMP are matching *if and only if* they are stable to SGD noise, and diagnosed this instability as a failure case of the original IMP algorithm (Frankle & Carbin, 2019). A simple treatment to ensure the stability is sharing the early phase of the optimization trajectory. In other words, there exists a linear connector between

$$\boldsymbol{w}_T^{(1)} \leftarrow \mathrm{SGD}_{T_0 \rightarrow T}(\boldsymbol{w}_{T_0}, \xi^{(1)}, \mathcal{D}) \text{ and } \boldsymbol{w}_T^{(2)} \leftarrow \mathrm{SGD}_{T_0 \rightarrow T}(\boldsymbol{w}_{T_0}, \xi^{(2)}, \mathcal{D}), \tag{2}$$

when SGD runs are started from the same initialization of $\boldsymbol{w}_{T_0} \leftarrow \mathrm{SGD}_{0 \rightarrow T_0}(\boldsymbol{w}_0, \xi, \mathcal{D})$. Furthermore, Paul et al. (2023) demonstrated linear connectivity between two consecutive IMP solutions with different sparsity levels and identified it as a crucial factor for the success of IMP.

Nevertheless, the question of whether a low-loss subspace is formed by the convex hull of three or more solutions remains uncertain, despite the presence of linear connectivity between each pair of solutions. If it becomes feasible to construct a low-loss volume subspace using IMP solutions, it could potentially yield a more effective solution with improved generalization at the midpoint of this subspace (Wortsman et al., 2021).

### 3.2 SWAMP: AN ALGORITHM

Inspired by the stability analysis of the matching ticket presented in § 3.1, we propose Sparse Weight Averaging with Multiple Particles (SWAMP) as a tailored sparse weight averaging technique for IMP. The detailed algorithm is presented in Algorithm 1.

SWAMP differs from vanilla IMP in two main aspects. Firstly, we create multiple copies of the matching ticket (line 5; Algorithm 1) and train them simultaneously with different random seeds

---

**Algorithm 1** Iterative Magnitude Pruning with SWAMP (ours)

---

**Require:** Neural network parameter $w$, pruning mask $m$, training dataset $\mathcal{D}$, the number of cycles for iterative magnitude pruning $C$, the number of iterations for each cycle $T$, pruning ratio $\alpha$, SGD noise $\xi$, the number of iteration for matching ticket $T_0$, and the number of particles $N$.
**Ensure:** Sparse solution $\overline{w}_{c,T}$.

---

1: Randomly initialize $w_{0,0}$ and set mask $m \leftarrow \mathbf{1}$.                    ▷ Starts from random dense weights.
2: Train $w_{0,T_0} \leftarrow \text{SGD}_{0 \rightarrow T_0}(w_{0,0} \circ m, \xi_0, \mathcal{D})$.    ▷ Gets a matching ticket from the initialization.
3: **for** $c \in \{1, \ldots, C\}$ **do**
4:     **for** $n \in \{1, \ldots, N\}$ **do**
5:         Rewind $w_{c,0}^{(n)} \leftarrow w_{0,T_0} \circ m$.                    ▷ Starts from the matching ticket.
6:         Train $w_{c,T}^{(n)} \leftarrow \text{SWA}_{0 \rightarrow T}(w_{c,0}^{(n)}, \xi_c^{(n)}, \mathcal{D})$.    ▷ Averages weights over trajectory.
7:     **end for**
8:     Averaging $\overline{w}_{c,T} \leftarrow \sum_{n=1}^{N} w_{c,T}^{(n)} / N$.                    ▷ Averages weights of particles.
9:     Prune $m \leftarrow \text{Prune}(\overline{w}_{c,T}, \alpha)$.                    ▷ Updates the mask based on magnitudes.
10: **end for**

---

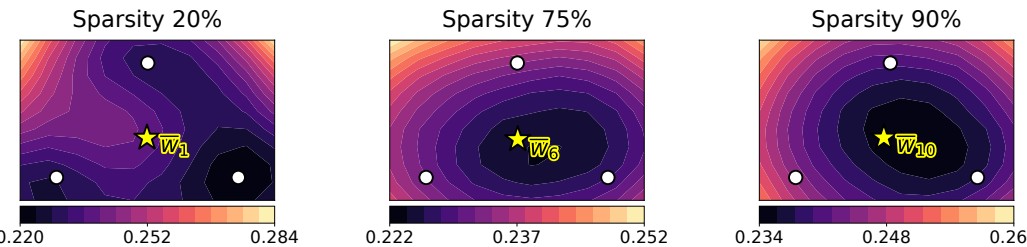

**Figure 2:** Visualization of loss surfaces as a function of network weights in a two-dimensional subspace, spanned by three particles (marked as white circles). The averaged weight $w_c$ (marked by a yellow star) is observed not to be positioned in the flat region of the surface during the earlier stages of IMP (left; Sparsity 20%). However, as sparsity increases, the weight averaging technique effectively captures the flat region of the surface. The results are presented for WRN-28-2 on the test split of CIFAR-10, and we refer the reader to Appendix C.8 for the same plot for CIFAR-100.

(line 6; Algorithm 1), whereas IMP employs a single particle. Secondly, we replace SGD training with Stochastic Weight Averaging (SWA; Izmailov et al., 2018), a method that constructs a moving average of parameters by periodically sampling a subset of the learning trajectory, and SWA enables us to accumulate virtually more particles throughout the training. We then average all particles before proceeding to the pruning step (line 8; Algorithm 1).

As illustrated in Figure 1, our algorithm achieves superior performance, which is on par with that of an ensemble consisting of two sparse networks. This is quite remarkable considering that our solution achieves this level of performance while having significantly lower inference costs compared to the ensembling approach. Further ablation studies presented in § 4.2 and Appendix C.4 also validate that both ingredients independently contribute to our algorithm, with each playing a crucial role in achieving superior performance.

### 3.3   SWAMP: A LOSS LANDSCAPE PERSPECTIVE

In this section, we explore step-by-step whether the characteristics of IMP introduced in § 3.1 also hold for SWAMP along with highlighting the strengths of SWAMP. To begin with, we examine the linear connectivity of SWAMP particles in a single cycle. Although Frankle et al. (2020) empirically proves pair-wise linear connectivity, it remains uncertain whether this holds true for the convex combination of more than two particles. In Figure 2, we visualize the loss surface of IMP-trained particles along with the weight-averaged particle. We can notice that weight averaging fails at the earlier stages of IMP due to the highly non-convex nature of the landscape. However, as sparsity increases, particles tend to locate in the same wide basin which enables weight-averaging. Such a

**Table 1:** Trace of Hessian $Tr(\mathbf{H})$ evaluated across training data. In most cases, SWAMP with multiple particles exhibits smaller trace value, i.e., finds flatter minima, compared to others. Reported values are averaged over three random seeds, and the best and second-best results are boldfaced and underlined, respectively. Refer to Appendix C.7 for the results on CIFAR-100.

| | | Sparsity | | | |
| --- | --- | --- | --- | --- | --- |
| | Training | 20% | 50% | 75% | 90% |
| CIFAR-10 (WRN-28-2) | SGD | 1784.64 $\pm$152.95 | 1704.69 $\pm$142.57 | 1719.49 $\pm$114.25 | 1710.28 $\pm$185.84 |
| | SWAMP ($N = 1$) | 500.24 $\pm$ 50.00 | 500.70 $\pm$ 21.57 | 543.28 $\pm$ 31.27 | 579.79 $\pm$ 20.69 |
| | SWAMP ($N = 4$) | **463.90** $\pm$ 34.84 | **467.45** $\pm$ 35.76 | **515.54** $\pm$ 20.23 | **533.62** $\pm$ 4.27 |
| CIFAR-10 (VGG-13) | SGD | 1155.64 $\pm$ 51.21 | 1231.69 $\pm$ 55.21 | 1143.70 $\pm$ 65.25 | 1197.73 $\pm$ 21.04 |
| | SWAMP ($N = 1$) | **427.92** $\pm$ 11.15 | 424.85 $\pm$ 6.84 | 450.44 $\pm$ 20.82 | 473.89 $\pm$ 55.69 |
| | SWAMP ($N = 4$) | 432.45 $\pm$ 28.87 | **403.32** $\pm$ 15.40 | **403.42** $\pm$ 40.56 | **449.36** $\pm$ 16.80 |

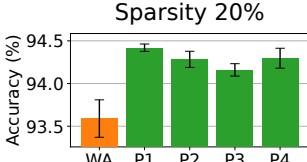 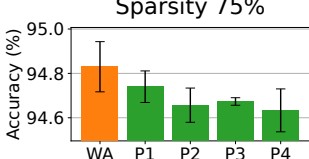 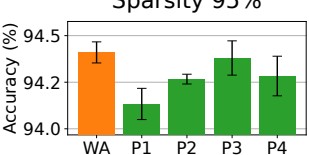

**Figure 3:** Bar plots depicting the accuracy of individual particles involved in the averaging process of the SWAMP algorithm. While the averaged weight (denoted as WA) may not outperform individual particles (denoted as P1-P4) in the early stages of IMP (left; Sparsity 20%), it achieves high performance at higher sparsity levels. The results are presented for WRN-28-2 on the test split of CIFAR-10. We refer readers to Appendix C.9 for the same plot for CIFAR-100.

finding is in line with Frankle et al. (2020) that demonstrated the ease of finding a low-loss curve with a smaller network compared to a larger one, i.e., a sparse network tends to be more stable. Additionally, it further demonstrates that our algorithm benefits more with sparser networks.

Figure 3 provides additional evidence showing that the weight-averaged solution is indeed superior to its individual members, other than in the cases where the dense network is not yet stabilized. Better, the notable performance gap between individual particles promotes the need for weight-averaging. We further quantify the flatness of the local minima through the trace of Hessian employing the Power Iteration algorithm (Yao et al., 2020). Higher Hessian trace value implies that the converged local minimum exhibits a high curvature. The results in Table 1 validate that SWAMP locates a flatter sparse network compared to IMP, only except for earlier cycles.

Finally, we check whether consecutive solutions from SWAMP cycles are linearly connected – a key to the success of IMP pointed out by Paul et al. (2023) – which indeed turns out to be true according to Figure 4. Not only is this true, but our method also exhibits minimal variance and maintains a highly stable trajectory throughout the pruning process, suggesting that SWAMP finds a flat and well-connected basin. To this end, we provide empirical evidence that our method effectively identifies flat minima while retaining the desirable properties of IMP, resulting in a single highly sparse network that outperforms IMP.

## 4 EXPERIMENTS

### 4.1 MAIN RESULTS: IMAGE CLASSIFICATION TASKS

**Baseline approaches.** In addition to IMP with weight rewinding (Frankle et al., 2020), our method is compared to a list of pruning techniques. This includes one-shot pruning methods (SNIP (Lee et al., 2019); GraSP (Wang et al., 2020); SynFlow (Tanaka et al., 2020)), dense-to-sparse training with dynamic masks (DST (Liu et al., 2020); RigL (Evci et al., 2020)), SAM-optimized IMP (Na et al., 2022), and Lottery Pools (Yin et al., 2023).

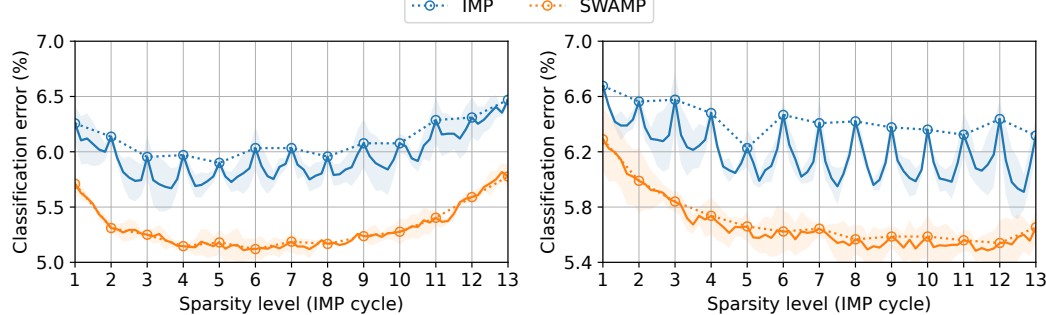

**Figure 4:** Linear connectivity between sparse solutions with different sparsity levels gathered from the end of IMP cycles. The results are presented for WRN-28-2 (left) and VGG-13 (right) on the test split of CIFAR-10, and we refer the reader to Appendix C.10 for the same plot for CIFAR-100.

**Table 2:** Classification accuracy on residual networks. SWAMP outperforms all the baselines across varying sparsities. Reported values are averaged over three random seeds, and the best and second-best results are boldfaced and underlined, respectively.

| | | Sparsity | | | |
|---|---|---|---|---|---|
| | Method | 50% | 75% | 90% | 95% |
| CIFAR-10 (WRN-28-2) | SNIP | $92.63_{\pm0.56}$ | $92.13_{\pm0.21}$ | $91.02_{\pm0.81}$ | $88.97_{\pm0.35}$ |
| | SynFlow | $92.99_{\pm0.46}$ | $92.98_{\pm0.08}$ | $90.98_{\pm0.30}$ | $89.53_{\pm0.20}$ |
| | GraSP | $92.34_{\pm0.26}$ | $90.74_{\pm0.27}$ | $89.70_{\pm0.60}$ | $88.78_{\pm0.22}$ |
| | RigL | $93.59_{\pm0.15}$ | $93.09_{\pm0.13}$ | $91.81_{\pm0.18}$ | $90.73_{\pm0.16}$ |
| | DST | $94.04_{\pm0.37}$ | $93.62_{\pm0.12}$ | $92.59_{\pm0.40}$ | $92.10_{\pm0.34}$ |
| | IMP | $93.97_{\pm0.16}$ | $94.02_{\pm0.23}$ | $93.90_{\pm0.15}$ | $\underline{93.58}_{\pm0.09}$ |
| | IMP + SAM | $94.06_{\pm0.30}$ | $94.20_{\pm0.54}$ | $93.89_{\pm0.53}$ | $92.03_{\pm1.93}$ |
| | Lottery Pools | $\underline{94.39}_{\pm0.16}$ | $\underline{94.28}_{\pm0.14}$ | $\underline{94.16}_{\pm0.11}$ | $93.43_{\pm0.23}$ |
| | **SWAMP (ours)** | $\mathbf{94.74}_{\pm0.04}$ | $\mathbf{94.88}_{\pm0.09}$ | $\mathbf{94.73}_{\pm0.10}$ | $\mathbf{94.23}_{\pm0.11}$ |
| CIFAR-100 (WRN-32-4) | SNIP | $71.98_{\pm0.37}$ | $72.16_{\pm0.77}$ | $69.82_{\pm1.14}$ | $68.10_{\pm0.28}$ |
| | SynFlow | $75.16_{\pm0.54}$ | $74.22_{\pm0.60}$ | $74.60_{\pm0.54}$ | $74.17_{\pm0.77}$ |
| | GraSP | $70.75_{\pm1.78}$ | $68.17_{\pm0.48}$ | $66.96_{\pm0.05}$ | $65.80_{\pm0.89}$ |
| | RigL | $75.05_{\pm0.48}$ | $73.37_{\pm0.11}$ | $71.23_{\pm0.76}$ | $70.23_{\pm0.48}$ |
| | DST | $74.78_{\pm1.23}$ | $74.02_{\pm1.73}$ | $72.79_{\pm1.16}$ | $71.41_{\pm0.43}$ |
| | IMP | $75.40_{\pm0.23}$ | $75.72_{\pm0.41}$ | $75.24_{\pm0.25}$ | $74.60_{\pm0.37}$ |
| | IMP + SAM | $75.63_{\pm0.70}$ | $\underline{76.19}_{\pm0.81}$ | $\underline{75.85}_{\pm0.78}$ | $75.07_{\pm0.65}$ |
| | Lottery Pools | $\underline{76.31}_{\pm0.51}$ | $76.17_{\pm1.03}$ | $75.84_{\pm0.67}$ | $\underline{75.14}_{\pm0.49}$ |
| | **SWAMP (ours)** | $\mathbf{77.29}_{\pm0.53}$ | $\mathbf{77.35}_{\pm0.39}$ | $\mathbf{77.14}_{\pm0.33}$ | $\mathbf{76.48}_{\pm0.73}$ |

**Experimental setup.** Our method is evaluated on diverse image classification benchmarks, which include CIFAR-10, CIFAR-100, Tiny-ImageNet and ImageNet datasets. Throughout the experiments, we use residual networks (He et al., 2016) and VGG networks (Simonyan & Zisserman, 2015) as a basis: WRN-28-2 and VGG-13 for CIFAR-10; WRN-32-4 and VGG-16 for CIFAR-100; R18 for Tiny-ImageNet; and R50 for ImageNet. Unless specified, we set the number of SWAMP particles $N = 4$ and pruning ratio $\alpha = 0.2$. Refer to Appendix B for further experimental details.

Table 2 presents the performance of SWAMP together with other baseline methods on the CIFAR-10 and CIFAR-100 datasets, respectively. Compared to other baseline methods, SWAMP consistently achieves the highest classification accuracy across all sparsity levels and models when evaluated on the CIFAR-10 and CIFAR-100 test sets; we defer results on VGG networks and Tiny-ImageNet to Appendix C. To see the uncertainty quantification aspects, in Appendix C.6, we report negative log-likelihoods (NLLs) as well. Again, SWAMP achieves the best NLL in all settings. Furthermore, Table 3 highlights that our method consistently outperforms IMP on ImageNet, a large-scale dataset that is known to be hard to prune.

**Table 3:** Classification accuracy with ResNet-50, which is trained on ImageNet-Train, validated on IN-Valid, and tested on ImageNet-V2, ImageNet-Rendition (denoted by IN-R), and ImageNet-Sketch (denoted by IN-S). Reported values are averaged over three trials.

| | Method | Sparsity | | | | | |
| | | 0% | 45.9% | 68.8% | 80.3% | 86.0% | 88.9 % |
|---|---|---|---|---|---|---|---|
| IN-Val | IMP | $76.25_{\pm0.04}$ | $76.40_{\pm0.10}$ | $76.13_{\pm0.09}$ | $75.54_{\pm0.07}$ | $74.22_{\pm0.09}$ | $71.66_{\pm0.09}$ |
| | **SWAMP** | - | $76.56_{\pm0.08}$ | $76.51_{\pm0.03}$ | $75.69_{\pm0.20}$ | $74.25_{\pm0.03}$ | $71.81_{\pm0.19}$ |
| IN-V2 | IMP | $64.18_{\pm0.10}$ | $64.11_{\pm0.12}$ | $63.78_{\pm0.03}$ | $62.77_{\pm0.19}$ | $61.20_{\pm0.19}$ | $59.00_{\pm0.32}$ |
| | **SWAMP** | - | $64.34_{\pm0.38}$ | $64.06_{\pm0.40}$ | $63.43_{\pm0.24}$ | $61.82_{\pm0.15}$ | $59.44_{\pm0.07}$ |
| IN-R | IMP | $35.38_{\pm0.27}$ | $35.04_{\pm0.25}$ | $34.71_{\pm0.19}$ | $34.05_{\pm0.26}$ | $32.85_{\pm0.29}$ | $30.92_{\pm0.10}$ |
| | **SWAMP** | - | $37.12_{\pm0.11}$ | $36.61_{\pm0.16}$ | $35.61_{\pm0.22}$ | $34.14_{\pm0.31}$ | $32.21_{\pm0.73}$ |
| IN-S | IMP | $23.90_{\pm0.02}$ | $23.86_{\pm0.23}$ | $23.74_{\pm0.32}$ | $22.88_{\pm0.26}$ | $21.60_{\pm0.24}$ | $19.32_{\pm0.46}$ |
| | **SWAMP** | - | $25.29_{\pm0.18}$ | $24.95_{\pm0.17}$ | $24.05_{\pm0.33}$ | $22.43_{\pm0.02}$ | $20.08_{\pm0.42}$ |

**Table 4:** Ablation study to validate SWAMP's efficacy in two aspects: (i) mask generation and (ii) sparse training. Except for earlier iterations, i.e., low sparsity regime, our method excels in both areas relative to vanilla SGD optimization. Reported values are classification accuracy averaged over three random seeds, and the best results are boldfaced. Refer to Appendix C.4 for VGG networks.

| | Mask | Training | Sparsity | | | |
| | | | 50% | 75% | 90% | 95% |
|---|---|---|---|---|---|---|
| CIFAR-10 (WRN-28-2) | SGD | SGD | $93.97_{\pm0.16}$ | $94.02_{\pm0.23}$ | $93.90_{\pm0.15}$ | $93.58_{\pm0.09}$ |
| | SGD | SWAMP | $\mathbf{94.85}_{\pm0.05}$ | $\mathbf{94.91}_{\pm0.09}$ | $94.48_{\pm0.06}$ | $93.99_{\pm0.31}$ |
| | SWAMP | SGD | $94.15_{\pm0.15}$ | $94.37_{\pm0.18}$ | $94.26_{\pm0.03}$ | $93.83_{\pm0.16}$ |
| | SWAMP | SWAMP | $94.74_{\pm0.04}$ | $94.88_{\pm0.09}$ | $\mathbf{94.73}_{\pm0.10}$ | $\mathbf{94.23}_{\pm0.11}$ |

## 4.2 ABLATION STUDIES

**Does SWAMP find a better mask?** To validate that SWAMP indeed identifies a superior mask compared to IMP, we conduct an experiment with two different masks: (i) a mask obtained from IMP, and (ii) a mask obtained from SWAMP. At a predetermined fixed sparsity level, we initially train our model using SGD (or SWAMP) and the mask from IMP. At the same time, we train the model using SGD (or SWAMP) and the mask from SWAMP at the same sparsity level. These two processes differ only in the masks utilized, while the training approach and the fixed sparsity level remain the same. Table 4 presents clear evidence that the SWAMP mask consistently outperforms its counterpart in terms of performance, with the exception of the WRN-28-2 model at 50% and 75% sparsity levels when trained using SWAMP. This result shows that SWAMP generates a *better* sparse mask than IMP.

**Does SWAMP offer a better sparse training?** In Table 4, we can also verify that SWAMP offers better sparse training compared to IMP. By comparing the results between SGD training and SWAMP training using the same mask, it becomes evident that SWAMP consistently outperforms across all masks, sparsity levels, and models. It verifies that SWAMP effectively guides the weight particles towards converging into flat local minima, resulting in improved generalization on the test split. The induced flatness of the local minima through SWAMP's weight distribution contributes to enhanced performance and robustness of the model as we discussed in § 3.3.

**Two averaging strategies of SWAMP.** As described in § 3.2, we here investigate how stochastic weight averaging and multi-particle averaging contribute to the final performance of SWAMP. In Table 5, throughout all four sparsities, applying only one of the two techniques displays better performance than IMP (bottom-row) but clearly lower than SWAMP (top-row). We conclude that two ingredients complement each other, achieving optimal performance when applied together. Further in Table 6, we conduct an empirical analysis to investigate the correlation between the number of particles and the performance of SWAMP. We provide additional ablation studies in Appendix C.4.

**Table 5:** Ablation study on the impact of the two main components of SWAMP; averaging multiple particles (denoted by Multi) and averaging learning trajectory (denoted by SWA). Our findings indicate that the best performance is achieved when both techniques are employed. Reported classification accuracies are averaged over three random seeds, and the best and second-best results are boldfaced and underlined, respectively. Refer to Appendix C.4 for VGG networks.

| | Multi | SWA | Sparsity | | | |
|---|---|---|---|---|---|---|
| | | | 50% | 75% | 90% | 95% |
| CIFAR-10 (WRN-28-2) | ✓ | ✓ | $\mathbf{94.74}_{\pm 0.04}$ | $\mathbf{94.88}_{\pm 0.09}$ | $\mathbf{94.73}_{\pm 0.10}$ | $\mathbf{94.23}_{\pm 0.11}$ |
| | ✓ | ✗ | $94.43_{\pm 0.10}$ | $94.44_{\pm 0.19}$ | $\underline{94.37}_{\pm 0.12}$ | $93.73_{\pm 0.32}$ |
| | ✗ | ✓ | $\underline{94.62}_{\pm 0.06}$ | $\underline{94.67}_{\pm 0.06}$ | $94.35_{\pm 0.06}$ | $\underline{93.97}_{\pm 0.10}$ |
| | ✗ | ✗ | $93.97_{\pm 0.16}$ | $94.02_{\pm 0.23}$ | $93.90_{\pm 0.15}$ | $93.58_{\pm 0.09}$ |

**Table 6:** Ablation study on the number of SWAMP particles. The finding indicates that the performance improves with an increase in the number of particles. Reported values are classification accuracy averaged over three random seeds, and the best and second-best results are boldfaced and underlined, respectively. Refer to Appendix C.4 for VGG networks.

| | # particles | Sparsity | | | |
|---|---|---|---|---|---|
| | | 50% | 75% | 90% | 95% |
| CIFAR-10 (WRN-28-2) | 1 | $94.62_{\pm 0.06}$ | $94.67_{\pm 0.06}$ | $94.35_{\pm 0.06}$ | $93.97_{\pm 0.10}$ |
| | 2 | $94.57_{\pm 0.04}$ | $94.59_{\pm 0.07}$ | $94.38_{\pm 0.17}$ | $94.05_{\pm 0.16}$ |
| | 4 | $\underline{94.74}_{\pm 0.04}$ | $\underline{94.88}_{\pm 0.09}$ | $\underline{94.73}_{\pm 0.10}$ | $\mathbf{94.23}_{\pm 0.11}$ |
| | 8 | $\mathbf{94.80}_{\pm 0.04}$ | $\mathbf{94.90}_{\pm 0.09}$ | $\mathbf{94.74}_{\pm 0.10}$ | $\underline{94.21}_{\pm 0.24}$ |

## 4.3 FURTHER REMARKS

**Parallelization strategy in distributed training.** A notable limitation of the SWAMP algorithm is its training cost, which scales linearly with the number of particles. The training cost is typically not a major issue when working with small datasets like CIFAR-10/100, but it becomes significant when handling large datasets like ImageNet. Consequently, we suggest parallelizing the particles across machines when implemented within distributed training environments, a common practice for handling large-scale models and datasets. This strategy incurs virtually no additional costs compared to IMP, except for the extra memory required for storing the averaged parameters. Indeed, we put this strategy into practice during our ImageNet experiments, and SWAMP achieved outstanding results while incurring almost the same training expenses as IMP (cf. Table 3).

**Reducing training costs of SWAMP.** The parallelization strategy mentioned above is exclusively applicable to distributed training setups. Consequently, we further introduce practical methods to reduce the training costs of the SWAMP algorithm, which can be used even in non-distributed training environments: *(1) Employing multiple particles only in the high-sparsity regime* mitigates the significant training costs mainly encountered in low-sparsity regimes. Table 7 demonstrates that this approach reduces training costs by a factor of 3 to 4 with minimal performance degradation. Here, we initiate training with a single-particle SWAMP for the first ten IMP cycles, achieving a sparsity level of 90%, and then transition to using four particles afterward. *(2) Increasing the pruning ratio* decreases the number of pruning cycles necessary to achieve a certain sparsity level and thus significantly reduces total training costs. Table 8 verifies that SWAMP is proficient in pruning even when using a higher pruning ratio, highlighting a distinctive advantage of SWAMP compared to IMP. These findings, combined with those in Table 1, align with Lemma 3.1 in Paul et al. (2023); a smaller Hessian eigenvalue, indicating flatter minima, enhances robustness to SGD noise and making it more likely to restore matching performance.

**Extension to language tasks and dynamic pruning.** Up to this point, we have demonstrated that multi-particle averaging benefits IMP in iamge classification tasks. However, it is important to note that SWAMP can be easily applied to different pruning techniques across a range of tasks. To clarify, we present two distinct extensions: (1) Table 9 further confirms that SWAMP outperforms IMP in language tasks, where experimental details are available in Appendix C.2. (2) We also illustrate how

**Table 7:** Further comparison between (i) IMP, (ii) SWAMP, and (iii) the cost-efficient version of the SWAMP algorithm (denoted by SWAMP+) in terms of accuracy and total training FLOPs. Reported values are averaged over three random seeds.

| | Method | Sparsity 95% | | Sparsity 98% | |
|---|---|---|---|---|---|
| | | Accuracy | GFLOPs | Accuracy | GFLOPs |
| CIFAR-10 (WRN-28-2) | IMP | $93.58_{\pm0.09}$ | 1.19 | $89.05_{\pm1.39}$ | 1.23 |
| | SWAMP | $94.23_{\pm0.11}$ | 4.75 | $90.85_{\pm0.47}$ | 4.92 |
| | SWAMP+ | $94.32_{\pm0.24}$ | 1.39 | $90.51_{\pm0.08}$ | 1.56 |
| CIFAR-100 (WRN-32-4) | IMP | $74.60_{\pm0.37}$ | 5.25 | $70.74_{\pm0.71}$ | 5.38 |
| | SWAMP | $76.48_{\pm0.73}$ | 20.99 | $72.14_{\pm0.58}$ | 21.53 |
| | SWAMP+ | $76.19_{\pm0.18}$ | 5.96 | $71.90_{\pm0.72}$ | 6.51 |

**Table 8:** Results on CIFAR for larger pruning ratio of $\alpha \in \{0.2, 0.3, 0.4, 0.5\}$ after 13, 8, 6, 4 IMP cycles, respectively. Reported values are averaged over three random seeds. The numbers within parentheses indicate the difference from the IMP baseline.

| | Method | $\alpha = 0.2$ | $\alpha = 0.3$ | $\alpha = 0.4$ | $\alpha = 0.5$ |
|---|---|---|---|---|---|
| CIFAR-10 (WRN-28-2) | IMP | $93.58_{\pm0.09}$ | $93.36_{\pm0.35}$ | $93.11_{\pm0.13}$ | $93.77_{\pm0.12}$ |
| | **SWAMP** | $94.23_{\pm0.11}$ (+0.65) | $93.83_{\pm0.04}$ (+0.47) | $94.41_{\pm0.02}$ (+1.30) | $94.68_{\pm0.04}$ (+0.91) |
| CIFAR-100 (WRN-32-4) | IMP | $74.60_{\pm0.37}$ | $74.24_{\pm0.33}$ | $73.87_{\pm0.86}$ | $73.85_{\pm0.83}$ |
| | **SWAMP** | $76.48_{\pm0.73}$ (+1.88) | $77.31_{\pm0.71}$ (+3.07) | $76.26_{\pm0.55}$ (+2.39) | $76.50_{\pm0.40}$ (+2.65) |

**Table 9:** F1 score and accuracy on the development set of MRPC and RTE, respectively. SWAMP displays improved performance on language tasks as well. Reported values are averaged over three random seeds, and the numbers within parentheses indicate the difference from the IMP baseline.

| | Method | Sparsity | | | |
|---|---|---|---|---|---|
| | | 0% | 40.9% | 52.2% | 61.3% |
| MRPC (RoBERTa) | IMP | $92.60_{\pm0.70}$ | $91.02_{\pm0.31}$ | $88.31_{\pm0.17}$ | $86.81_{\pm0.50}$ |
| | **SWAMP** | $92.81_{\pm0.38}$ (+0.21) | $91.51_{\pm0.39}$ (+0.49) | $89.40_{\pm0.40}$ (+1.09) | $88.12_{\pm0.48}$ (+1.31) |
| RTE (RoBERTa) | IMP | $80.13_{\pm1.02}$ | $73.25_{\pm1.52}$ | $63.30_{\pm0.37}$ | $57.46_{\pm1.39}$ |
| | **SWAMP** | $80.30_{\pm0.19}$ (+0.17) | $75.40_{\pm1.45}$ (+2.15) | $68.91_{\pm1.08}$ (+5.61) | $62.63_{\pm1.30}$ (+5.17) |

SWAMP has the potential to enhance dynamic sparse training methods, along with additional results with RigL (Evci et al., 2020) in Appendix C.3.

## 5 CONCLUSION

Drawing inspiration from previous research on the relationship between iterative magnitude pruning and linear mode connectivity, we extended the single-particle scenario to incorporate multiple particles. Our initial empirical findings demonstrated that multiple models trained with different SGD noise but sharing the same matching ticket can be weight-averaged without encountering loss barriers. We further observed that the averaged particle results in flat minima with improved generalization performance. In light of these insights, we introduced SWAMP, a novel iterative global pruning technique. We also established that SWAMP preserves the linear connectivity between consecutive solutions, a critical factor contributing to the effectiveness of IMP. Extensive experiments showed that SWAMP generates superior sparse masks and effectively trains sparse networks over other baseline methods. A theoretical analysis exploring why the convex hull of the particles in weight space forms a low-loss subspace would be a valuable direction for future research. Further, investigating the underlying principles and mathematical properties of the convex hull of the solution particles and its relationship to the low-loss subspace could provide insights into the behavior and effectiveness of the SWAMP algorithm.

ACKNOWLEDGEMENT

This work was supported by Institute of Information & communications Technology Planning & Evaluation (IITP) grant funded by the Korea government (MSIT) (No.2019-0-00075, Artificial Intelligence Graduate School Program (KAIST), No.2022-0-00713, Meta-learning Applicable to Real-world Problems, No.2022-0-00184, Development and Study of AI Technologies to Inexpensively Conform to Evolving Policy on Ethics), and the National Research Foundation of Korea (NRF) grant funded by the Korea government (MSIT) (NRF-2022R1A5A708390812).

REPRODUCIBILITY STATEMENT

Our algorithm is built on Pytorch 1.10.2 (Paszke et al., 2019), which is available under a BSD-style license[1]. All experiments are conducted on NVIDIA RTX 2080 and NVIDIA RTX 3090 machines. Basic experimental setup including network and dataset choice is listed in § 4.1. In Appendix B, we further provide all detailed hyperparameter settings to ensure the fair comparison between the baselines and our algorithm. We also include our code in the supplementary material.

ETHICS STATEMENT

The paper does not raise any ethical concerns. We only utilize publicly available datasets and python packages adhering to the appropriate licenses.

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

## A    RELATED WORKS

**Lottery ticket hypothesis.**   Neural network pruning techniques aim to identify parameters that can be removed from the network without affecting its performance. The simplest yet effective pruning strategy is *magnitude-based pruning*, which involves removing parameters with small magnitudes (Janowsky, 1989; Han et al., 2015). While there is no guarantee that zeroing the weights that are close to zero will mitigate the increase in training loss (LeCun et al., 1989; Hassibi & Stork, 1992), magnitude-based pruning can still be effective as it minimizes changes in the output of each layer (Dong et al., 2017; Park et al., 2020). Empirical studies have demonstrated that *iterative magnitude pruning (IMP)* - applying magnitude-based pruning multiple times - can effectively remove a large proportion of weights in neural networks (Frankle & Carbin, 2019; Renda et al., 2020; Frankle et al., 2020).

A large body of work tries to unravel the success of lottery tickets given the increasing interest across many applications (Chen et al., 2021; Kalibhat et al., 2021; Girish et al., 2021). In light of existing sanity-checking methods such as score inversion and layer-wise random shuffling, IMP passes those sanity checks while other one-shot pruning methods fail, which shows the matching tickets do contain useful information regarding network weights (Ma et al., 2021; Frankle et al., 2021; Su et al., 2020). Zhou et al. (2019) focus on the *z*eroing out weights explaining such operation behaves resembles neural network training along with putting emphasis on preserving the sign of weights. Renda et al. (2020) proposes learning rate rewinding which is empirically shown to achieve superior performance compared to weight rewinding and fine-tuning. Inspired by Gordon's escape theorem, Larsen et al. (2022) theoretically relates the matching ticket to a burn-in random subspace crossing the loss sublevel set.

**More pruning literature.**   Lottery tickets are primarily hindered by their substantial training costs whereas there exist one-shot pruning methods which only require cost corresponding to a single cycle of IMP. As a pioneering work, Lee et al. (2019) proposes SNIP which approximates the sensitivity of a connection, i.e., the impact of removing a single connection. GraSP (Wang et al., 2020) suggests preserving the gradient flow as a pruning criterion, and SynFlow (Tanaka et al., 2020) avoids the layer-collapse phenomenon without looking at the training data. With the intention of catching up with such computationally efficient algorithms, You et al. (2020) proposes early-bird tickets which significantly reduces the cost of the mask-searching step. As IMP provides so-called sparse-to-sparse training, i.e., the mask is pre-defined, some work investigates dense-to-sparse training with dynamic masks (Evci et al., 2020; Liu et al., 2020). Liu et al. (2020) employs a trainable mask layer jointly optimizing masking threshold and weight throughout a single cycle. Evci et al. (2020) proposes a rigged lottery (RigL) that regrows the dead weights yet with a large gradient flow. It has been demonstrated empirically that RigL can escape local minima and discover improved basins during optimization process.

**Loss landscape and weight averaging.**   Recent studies have shown that the loss landscape of modern deep neural networks exhibits *mode-connectivity*, which means that the local minima found through stochastic optimization are connected by continuous low-loss paths (Draxler et al., 2018; Garipov et al., 2018). It motivates researchers to explore subspaces in the weight space that contain high-performing solutions (Izmailov et al., 2020; Wortsman et al., 2021). One promising approach to obtain such solutions is *weight averaging*, which involves taking the average of weights across multiple points (Izmailov et al., 2018; Wortsman et al., 2022). The main assumption of the weight averaging strategy is that all points are located in the same basin of attraction, which may not hold in the case of sparse networks where different sparse structures may correspond to different areas in the loss landscape. However, recent works (Frankle et al., 2020; Evci et al., 2022; Paul et al., 2023) suggest that this assumption may hold even for sparse networks, as there is a linear connectivity between the solutions of different sparsities obtained through iterative magnitude pruning. Frankle et al. (2020) show that two neural networks with the same matching ticket yet trained with different SGD noises are linearly connected without a high loss barrier in between. Evci et al. (2022) argues that lottery tickets are biased towards the final pruned solution, i.e., winning tickets enables relearning the previous solution. Paul et al. (2023) identify the existence of a low-loss curve connecting two solutions from consecutive IMP cycles, which guarantees that SGD noise cannot significantly alter the optimization path. Recently, Yin et al. (2023) propose to weight-average the tickets generated from consecutive IMP iterations in order to obtain a single stronger sparse network.

## B  EXPERIMENTAL DETAILS

Our algorithm is built on Pytorch 1.10.2 (Paszke et al., 2019), which is available under a BSD-style license[2]. All experiments are conducted on NVIDIA RTX 2080 and NVIDIA RTX 3090 machines.

**Pruning.**  Throughout our experiments, we exclusively perform weight pruning on the convolutional layer, leaving the batch normalization and fully-connected layers unpruned. Additionally, we do not utilize any predefined layer-wise pruning ratio. It is worth noting that we empirically checked that including the fully-connected layers as part of the prunable parameters does not affect the results we have obtained. By default, we use the constant pruning ratio of $\alpha = 0.2$ at all IMP pruning rounds, i.e., we retain 80% of weight. For the convenience of the readers, we provide approximate sparsity values at the 3rd, 6th, 10th, 13th cycles as 50%, 75%, 90%, and 95%, respectively, while the precise sparsity values are 48.80%, 73.79%, 89.26%, and 94.50%. Unless otherwise specified, we set the number of SWAMP particles to four.

**Baselines.**  For the fair comparison in terms of training epochs, we train all the baselines listed in § 4.1 until convergence. Merely increasing the number of training epochs does not benefit baselines; the performance of SWAMP cannot be pertained to its extensive training computations. For one-shot pruning methods (Lee et al., 2019; Wang et al., 2020; Tanaka et al., 2020), we post-train for 300 epochs after obtaining the sparse mask, and follow the same hyperparameter setting in the original paper. For DST (Liu et al., 2020), we explore $\alpha \in [1e-7, 1e-5]$ for all models and datasets to match similar sparsity levels with other baselines. For RigL (Evci et al., 2020), we search $\alpha \in [0.3, 0.5, 0.7]$. Here we fixed hyperparameter $\delta = 100$. We explore $\rho \in \{0.005, 0.01, 0.05, 0.1, 0.5\}$ for SAM (Na et al., 2022) with the same hyperparameter and optimizer setting of SWAMP. For Lottery Pools (Yin et al., 2023), we explore $\alpha \in \{0.01, 0.05, 0.1, 0.2, 0.3, 0.4, 0.5, 0.6, 0.7, 0.8, 0.9, 0.99\}$, and greedily choose among 7 candidate models w.r.t. validation accuracy.

**Optimization.**  We utilize the SGD optimizer with a momentum of 0.9 and a learning rate schedule that follows the cosine-decaying method. We choose the cosine-decaying schedule as it yields better results compared to the step decay schedule commonly used in lottery ticket literature.

**CIFAR-10/100.**  To obtain a matching solution, we first train the model for 10 epochs using a constant learning rate of 0.1, a weight decay 1e-4, and a batch size of 128. Subsequently, we employ a cosine decaying learning rate schedule over 150 training epochs in each cycle of the IMP algorithm. During the final quarter of the training epochs, we collect particles for SWAMP, resulting in a total of 38 particles in a single trajectory. The learning rate for this phase is set to a constant value of 0.05.

**Tiny-ImageNet.**  To obtain a matching solution, we first train the model for 20 epochs using a constant learning rate of 0.1, a weight decay 5e-4, and a batch size of 256. Subsequently, we employ a cosine decaying learning rate schedule over 160 training epochs in each cycle of the IMP algorithm. During the final quarter of the training epochs, we collect particles for SWAMP, resulting in a total of 40 particles in a single trajectory. The learning rate for this phase is set to a constant value of 0.05.

**ImageNet.**  To obtain a matching solution, we first train the model for 30 epochs using a constant learning rate of 0.8, a weight decay 1e-4, and a batch size of 2048. Subsequently, we employ a cosine decaying learning rate schedule over 60 training epochs in each cycle of the IMP algorithm. During the final quarter of the training epochs, we collect particles for SWAMP, resulting in a total of 15 particles in a single trajectory. The learning rate for this phase is set to a constant value of 0.004. We establish a distributed training environment comprising eight machines, and consequently, we distribute particles across these eight machines as discussed in § 4.2. We also employed the *learning rate rewinding* (Renda et al., 2020) instead of *weight rewinding* in ImageNet experiments.

---

[2]https://github.com/pytorch/pytorch/blob/main/LICENSE

---

**Algorithm 2** Stochastic Gradient Descent (i.e., $\text{SGD}_{0 \to T}(\boldsymbol{w}_0, \xi, \mathcal{D})$ )

---

**Require:** An initial neural network parameter $\boldsymbol{w}_0$, training dataset $\mathcal{D}$, the number of training iteration $T$, a learning rate $\eta$, a loss function $\mathcal{L}$, and SGD noise $\xi$.
**Ensure:** Solution $\boldsymbol{w}_T$.

1: **for** $t \in \{1, \dots, T\}$ **do**
2:      Sample a mini-batch $\mathcal{B} \subset \mathcal{D}$.
3:      Update parameter $\boldsymbol{w}_t \leftarrow \boldsymbol{w}_{t-1} - \eta \cdot \nabla_{\boldsymbol{w}} \mathcal{L}(\boldsymbol{w}; \mathcal{B})$.
4: **end for**

---

**Algorithm 3** Stochastic Weight Averaging (i.e., $\text{SWA}_{0 \to T}(\boldsymbol{w}_0, \xi, \mathcal{D})$ )

---

**Require:** An initial neural network parameter $\boldsymbol{w}_0$, training dataset $\mathcal{D}$, the number of training iteration $T$, a learning rate $\eta$, a loss function $\mathcal{L}$, and SGD noise $\xi$.
**Ensure:** Solution $\boldsymbol{w}_{\text{SWA}}$.

1: Initialize $\boldsymbol{w}_{\text{SWA}} \leftarrow \boldsymbol{0}$ and $n_{\text{SWA}} \leftarrow 0$.
2: **for** $t \in \{1, \dots, T\}$ **do**
3:      Sample a mini-batch $\mathcal{B} \subset \mathcal{D}$.
4:      Update parameter $\boldsymbol{w}_t \leftarrow \boldsymbol{w}_{t-1} - \eta \cdot \nabla_{\boldsymbol{w}} \mathcal{L}(\boldsymbol{w}; \mathcal{B})$.
5:      Periodically update $\boldsymbol{w}_{\text{SWA}} \leftarrow (n_{\text{SWA}} \boldsymbol{w}_{\text{SWA}} + \boldsymbol{w}_t)/(n+1)$ and $n_{\text{SWA}} \leftarrow n_{\text{SWA}} + 1$.
6: **end for**

---

## C   SUPPLEMENTARY RESULTS

### C.1   ALGORITHMS

**Stochastic Weight Averaging**   Algorithms 2 and 3 describe a detailed procedure for stochastic gradient descent (SGD; Robbins & Monro, 1951) and stochastic weight averaging (SWA; Izmailov et al., 2018), which are respectively denoted as $\text{SGD}_{0 \to T}(\boldsymbol{w}_0, \xi, \mathcal{D})$ and $\text{SWA}_{0 \to T}(\boldsymbol{w}_0, \xi, \mathcal{D})$ in the main text of the paper for notational simplicity. For SWA, we took a moving average over model copies sampled from the last 25% of the training epochs.

**Iterative Magnitude Pruning**   Algorithm 4 and Algorithm 5 respectively provide a detailed procedure for the vanilla iterative magnitude pruning (IMP; Frankle & Carbin, 2019) and IMP with weight rewinding (IMP-WR; Frankle et al., 2020). While IMP rewinds the network to the initialization after each cycle, IMP-WR rewinds the weights to the early epoch of the training process. Prune$(\boldsymbol{w}, \alpha)$ returns a pruning mask where $(1 - \alpha)$ of the remaining parameters are pruned based on their magnitudes.

### C.2   EXTENSION TO LANGUAGE TASKS

Table 9 verified the capability of SWAMP to language tasks. Specifically, we fine-tuned the *RoBERTa-Base* model (Liu et al., 2019) on two subtasks from GLUE benchmark (Wang et al., 2019): Microsoft Research Paraphrase Corpus (MRPC; Dolan & Brockett, 2005) and Recognizing Textual Entailment (RTE; Dagan et al., 2006; Bar Haim et al., 2006; Giampiccolo et al., 2007; Bentivogli et al., 2009). We mainly followed the experimental configuration outlined in the work of Liu et al. (2019), which includes specifications such as a token limit of 512, and utilization of the Adam optimizer (Kingma & Ba, 2015) with first-moment coefficient set to 0.9 and second-moment coefficient set to 0.98.

To obtain a matching solution, we first train the model for 5 epochs using a constant learning rate of 1e-05. Subsequently, we employ a linear decaying learning rate schdule over 10 training epochs in each cycle of the IMP algorithm. For SWAMP. we collect particles distributed across eight machines during the final half of the training epochs. Figure 5 summarizes the results for each IMP cycle, where we used the pruning ratio of $\alpha = 0.9$. We can readily find out that SWAMP outperforms the IMP baseline by a noticeable margin on both datasets.

---

**Algorithm 4** Iterative Magnitude Pruning (Frankle & Carbin, 2019)

---

**Require:** Neural network parameter $w$, pruning mask $m$, training dataset $\mathcal{D}$, the number of cycles for iterative magnitude pruning $C$, the number of iterations for each cycle $T$, pruning ratio $\alpha$, and SGD noise $\xi$.
**Ensure:** Sparse solution $w_{c,T}$.

1: Randomly initialize $w_{0,0}$ and set mask $m \leftarrow 1$.      ▷ Starts from random dense weights.
2: **for** $c \in \{1, \ldots, C\}$ **do**
3:      Reset $w_{c,0} \leftarrow w_{0,T_0} \circ m$.      ▷ Starts from the matching ticket.
4:      Train $w_{c,T} \leftarrow \text{SGD}_{0 \to T}(w_{c,0}, \xi_c, \mathcal{D})$.      ▷ Averages weights over trajectory.
5:      Prune $m \leftarrow \text{Prune}(w_{c,T}, \alpha)$.      ▷ Updates the mask based on magnitudes.
6: **end for**

---

**Algorithm 5** Iterative Magnitude Pruning with Rewinding (Frankle et al., 2020)

---

**Require:** Neural network parameter $w$, pruning mask $m$, training dataset $\mathcal{D}$, the number of cycles for iterative magnitude pruning $C$, the number of iterations for each cycle $T$, pruning ratio $\alpha$, SGD noise $\xi$, and the number of iteration for matching ticket $T_0$.
**Ensure:** Sparse solution $w_{c,T}$.

1: Randomly initialize $w_{0,0}$ and set mask $m \leftarrow 1$.      ▷ Starts from random dense weights.
2: Train $w_{0,T_0} \leftarrow \text{SGD}_{0 \to T_0}(w_{0,0} \circ m, \xi_0, \mathcal{D})$.      ▷ Gets a matching ticket from the initialization.
3: **for** $c \in \{1, \ldots, C\}$ **do**
4:      Rewind $w_{c,0} \leftarrow w_{0,T_0} \circ m$.      ▷ Starts from the matching ticket.
5:      Train $w_{c,T} \leftarrow \text{SGD}_{0 \to T}(w_{c,0}, \xi_c, \mathcal{D})$.      ▷ Averages weights over trajectory.
6:      Prune $m \leftarrow \text{Prune}(w_{c,T}, \alpha)$.      ▷ Updates the mask based on magnitudes.
7: **end for**

---

### C.3    EXTENSION TO DYNAMIC PRUNING METHODS

In this paper, we extend IMP to SWAMP leveraging particle averaging heavily inspired by Paul et al. (2023). Although the primary focus of our paper is the loss landscape analysis of IMP, one can readily apply SWAMP to dynamic pruning methods as well. Here we further extend our algorithm to encompass dense-to-sparse training, offering a means to mitigate the intensive training costs associated with IMP which includes multiple rounds of retraining. Our approach enhances RigL (Evci et al., 2020) by introducing multiple particles during the final training phase, integrated after completing 75% of the total training epochs. The results presented in Table 10 demonstrate that our method gracefully merges with RigL, incurring only marginal training overhead. This result shows that SWAMP can be naturally applied to dynamic pruning methods.

### C.4    FURTHER ABLATION STUDIES

As described in § 3.2, we conduct an empirical analysis to investigate the correlation between the number of particles and the performance of SWAMP. Table 11 presents our findings, demonstrating a strong positive correlation between the generalization performance of SWAMP and the number of individual particles. These results indicate that maximizing the number of particles used in SWAMP can significantly enhance model performance. Additionally, they provide evidence supporting the mention made in § 3.2 that both SWA and particle averaging contribute independently and complementarily to the overall effectiveness of the SWAMP algorithm. We also provide additional ablation results with VGG networks in Table 12 and Table 13.

### C.5    FURTHER RESULTS ON TINY-IMAGENET

Figure 6 depicts the results on Tiny-ImageNet. In the case of ResNet-18 on Tiny-ImageNet, SWAMP outperforms the ensemble performance of two sparse networks.

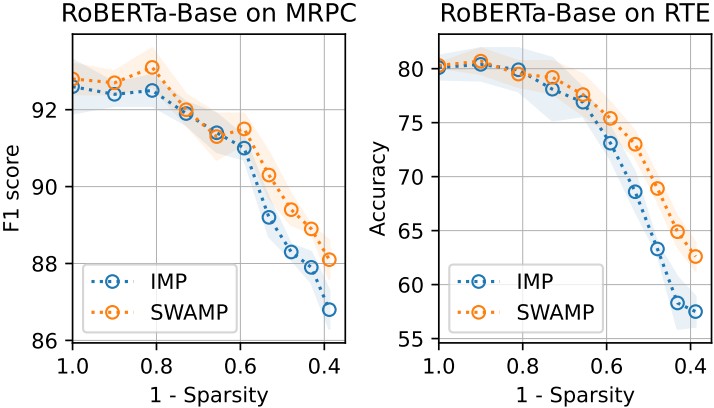

**Figure 5:** Results for RoBERT on MRPC and RTE. Reported values are averaged over three random seeds.

**Table 10:** Further comparison between (a) RigL and (b) RigL + SWAMP on CIFAR-10. Reported values are averaged over three random seeds.

| WRN28x2 | Sparsity | | |
|---|---|---|---|
| | 75% | 90% | 95% |
| (a) | $93.09_{\pm0.13}$ | $91.81_{\pm0.18}$ | $90.73_{\pm0.16}$ |
| (b) | $\mathbf{93.41}_{\pm0.04}$ | $\mathbf{92.78}_{\pm0.10}$ | $\mathbf{92.40}_{\pm0.46}$ |

| VGG-13 | Sparsity | | |
|---|---|---|---|
| | 75% | 90% | 95% |
| (a) | $93.18_{\pm0.04}$ | $92.46_{\pm0.16}$ | $91.45_{\pm0.16}$ |
| (b) | $\mathbf{93.53}_{\pm0.08}$ | $\mathbf{93.01}_{\pm0.16}$ | $\mathbf{93.01}_{\pm0.23}$ |

### C.6 SPARSITY PLOTS

We provide supplementary results in Figure 7, which clearly indicate that SWAMP performs better than IMP in terms of overall performance. Furthermore, we report the negative log-likelihood (NLL) values in Tables 15 and 16, where we employed temperature scaling (Guo et al., 2017) for better in-domain uncertainty evaluation, as discussed in Ashukha et al. (2020). Notably, SWAMP consistently outperforms other baselines in terms of NLL as well.

### C.7 FLATNESS OF LOCAL MINIMA

We further provide the results on the trace of Hessian for CIFAR-100 in Table 17 to show SWAMP with multiple particles finds flatter local minima than IMP leading to a better generalization performance.

### C.8 VISUALIZATION OF LOSS SURFACES

We provide additional loss landscape visualization plots for CIFAR-100 using WRN-32-4 model in Figure 8. We can observe a similar trend that SWAMP averages better as network sparsity grows.

### C.9 PARTICLE-WISE PERFORMANCE

We provide the particle-wise accuracies for CIFAR-100 in Figure 9. Individual SWAMP particles outperform the weight-averaged particle in lower sparsities, but we can observe the opposite trend in higher sparsities which is the area of our particular interest.

**Table 11:** Ablation study on the number of SWAMP particles. Performance improves with an increase in the number of particles. Reported values are classification accuracy averaged over three random seeds, and the best and second-best results are boldfaced and underlined, respectively.

|  | # particles | Sparsity | | | |
| --- | --- | --- | --- | --- | --- |
|  |  | 50% | 75% | 90% | 95% |
| CIFAR-10 (WRN-28-2) | 1 | $94.62 \pm 0.06$ | $94.67 \pm 0.06$ | $94.35 \pm 0.06$ | $93.97 \pm 0.10$ |
|  | 2 | $94.57 \pm 0.04$ | $94.59 \pm 0.07$ | $94.38 \pm 0.17$ | $94.05 \pm 0.16$ |
|  | 4 | $\underline{94.74} \pm 0.04$ | $\underline{94.88} \pm 0.09$ | $\underline{94.73} \pm 0.10$ | $\mathbf{94.23 \pm 0.11}$ |
|  | 8 | $\mathbf{94.80 \pm 0.04}$ | $\mathbf{94.90 \pm 0.09}$ | $\mathbf{94.74 \pm 0.10}$ | $\underline{94.21} \pm 0.24$ |
| CIFAR-10 (VGG-13) | 1 | $94.07 \pm 0.03$ | $94.15 \pm 0.09$ | $94.12 \pm 0.22$ | $93.99 \pm 0.17$ |
|  | 2 | $94.03 \pm 0.10$ | $93.93 \pm 0.16$ | $94.01 \pm 0.09$ | $94.03 \pm 0.07$ |
|  | 4 | $\underline{94.14} \pm 0.08$ | $\underline{94.39} \pm 0.15$ | $\underline{94.40} \pm 0.16$ | $\underline{94.34} \pm 0.11$ |
|  | 8 | $\mathbf{94.24 \pm 0.18}$ | $\mathbf{94.41 \pm 0.09}$ | $\mathbf{94.54 \pm 0.09}$ | $\mathbf{94.41 \pm 0.15}$ |

**Table 12:** Ablation study to validate SWAMP's efficacy in two aspects: (i) mask generation and (ii) sparse training. Except for earlier iterations, i.e., low sparsity regime, our method excels in both areas relative to vanilla SGD optimization. Reported values are classification accuracy averaged over three random seeds, and the best results are boldfaced.

|  | Mask | Training | Sparsity | | | |
| --- | --- | --- | --- | --- | --- | --- |
|  |  |  | 50% | 75% | 90% | 95% |
| CIFAR-10 (VGG-13) | SGD | SGD | $93.34_{\pm 0.17}$ | $93.53_{\pm 0.15}$ | $93.58_{\pm 0.03}$ | $93.62_{\pm 0.11}$ |
|  | SGD | SWAMP | $94.01_{\pm 0.02}$ | $94.30_{\pm 0.01}$ | $94.35_{\pm 0.08}$ | $94.18_{\pm 0.13}$ |
|  | SWAMP | SGD | $93.55_{\pm 0.03}$ | $93.84_{\pm 0.10}$ | $94.07_{\pm 0.07}$ | $93.98_{\pm 0.07}$ |
|  | SWAMP | SWAMP | $\mathbf{94.14}_{\pm 0.08}$ | $\mathbf{94.39}_{\pm 0.15}$ | $\mathbf{94.40}_{\pm 0.16}$ | $\mathbf{94.34}_{\pm 0.11}$ |

## C.10 CONNECTIVITY BETWEEN SUCCESSIVE SWAMP SOLUTIONS

We present additional results on CIFAR-100 regarding the linear connectivity between two consecutive SWAMP solutions. Following Paul et al. (2023), we find that SWAMP solutions from different cycles are also linearly well-connected.

**Table 13:** Ablation study on the impact of the two main components of SWAMP; averaging multiple particles (denoted by Multi) and averaging learning trajectory (denoted by SWA). Our findings indicate that the best performance is achieved when both techniques are employed. Reported classification accuracies are averaged over three random seeds, and the best and second-best results are boldfaced and underlined, respectively.

| | Multi | SWA | Sparsity | | | |
|---|---|---|---|---|---|---|
| | | | 50% | 75% | 90% | 95% |
| CIFAR-10 (VGG-13) | ✓ | ✓ | **94.14**$_{\pm 0.08}$ | **94.39**$_{\pm 0.15}$ | **94.40**$_{\pm 0.16}$ | **94.34**$_{\pm 0.11}$ |
| | ✓ | ✗ | 93.96$_{\pm 0.03}$ | 94.09$_{\pm 0.09}$ | 94.08$_{\pm 0.18}$ | 94.05$_{\pm 0.11}$ |
| | ✗ | ✓ | 94.07$_{\pm 0.03}$ | 94.15$_{\pm 0.09}$ | 94.12$_{\pm 0.22}$ | 93.99$_{\pm 0.17}$ |
| | ✗ | ✗ | 93.34$_{\pm 0.17}$ | 93.53$_{\pm 0.15}$ | 93.58$_{\pm 0.03}$ | 93.62$_{\pm 0.11}$ |

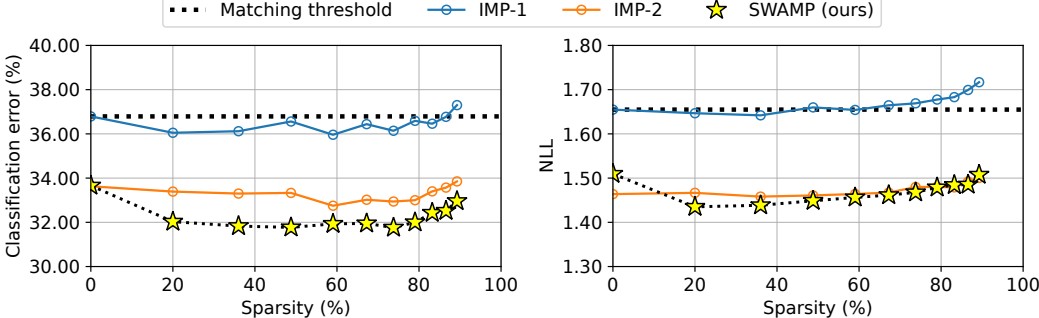

**Figure 6:** Classification error (left) and negative log-likelihood as a function of the sparsity. SWAMP achieves remarkable performance comparable to an ensemble of IMP solutions while maintaining the same inference cost as a single model. Here $n$ in the notation IMP-$n$ indicates the number of IMP ensemble members. Notably, SWAMP demonstrates *matching* performance even at extremely sparse levels, unlike IMP. The results are presented for ResNet-18 on Tiny-ImageNet.

**Table 14:** Classification accuracy on VGG networks. SWAMP outperforms all the baselines across varying sparsities, including one-shot and dynamic pruning approaches. Reported values are averaged over three random seeds, and the best and second-best results are boldfaced and underlined, respectively.

| | Method | Sparsity | | | |
|---|---|---|---|---|---|
| | | 50% | 75% | 90% | 95% |
| CIFAR-10 (VGG-13) | SNIP | 92.85 ± 0.19 | 92.59 ± 0.22 | 91.30 ± 0.26 | 90.34 ± 0.25 |
| | SynFlow | 93.01 ± 0.27 | 93.09 ± 0.27 | 92.74 ± 0.40 | 91.54 ± 0.21 |
| | GraSP | 92.20 ± 0.06 | 91.94 ± 0.27 | 91.16 ± 0.11 | 90.68 ± 0.08 |
| | RigL | 93.56 ± 0.18 | 93.18 ± 0.04 | 92.46 ± 0.16 | 91.45 ± 0.16 |
| | DST | 93.93 ± 0.20 | 93.90 ± 0.06 | 93.75 ± 0.02 | 93.38 ± 0.06 |
| | IMP | 93.34 ± 0.17 | 93.53 ± 0.15 | 93.58 ± 0.03 | 93.62 ± 0.11 |
| | IMP + SAM | 93.73 ± 0.12 | 93.96 ± 0.19 | 94.01 ± 0.12 | 93.89 ± 0.13 |
| | **SWAMP (ours)** | **94.14 ± 0.08** | **94.39 ± 0.15** | **94.40 ± 0.16** | **94.34 ± 0.11** |
| CIFAR-100 (VGG-16) | SNIP | 71.21 ± 0.21 | 70.41 ± 0.31 | 67.96 ± 0.15 | 66.24 ± 0.40 |
| | SynFlow | 71.52 ± 0.05 | 71.31 ± 0.13 | 71.00 ± 0.22 | 67.18 ± 0.24 |
| | GraSP | 69.08 ± 0.25 | 67.26 ± 0.06 | 65.25 ± 0.38 | 63.50 ± 0.09 |
| | RigL | 72.22 ± 0.43 | 71.38 ± 0.07 | 69.59 ± 0.41 | 67.44 ± 0.05 |
| | DST | 72.93 ± 0.16 | 72.87 ± 0.13 | 72.45 ± 0.10 | 71.52 ± 0.12 |
| | IMP | 72.32 ± 0.15 | 72.50 ± 0.15 | 72.64 ± 0.19 | 72.74 ± 0.15 |
| | IMP + SAM | 72.11 ± 0.45 | 72.27 ± 0.12 | 72.56 ± 0.19 | 72.50 ± 0.04 |
| | **SWAMP (ours)** | **73.27 ± 0.26** | **73.54 ± 0.36** | **73.40 ± 0.33** | **73.53 ± 0.32** |

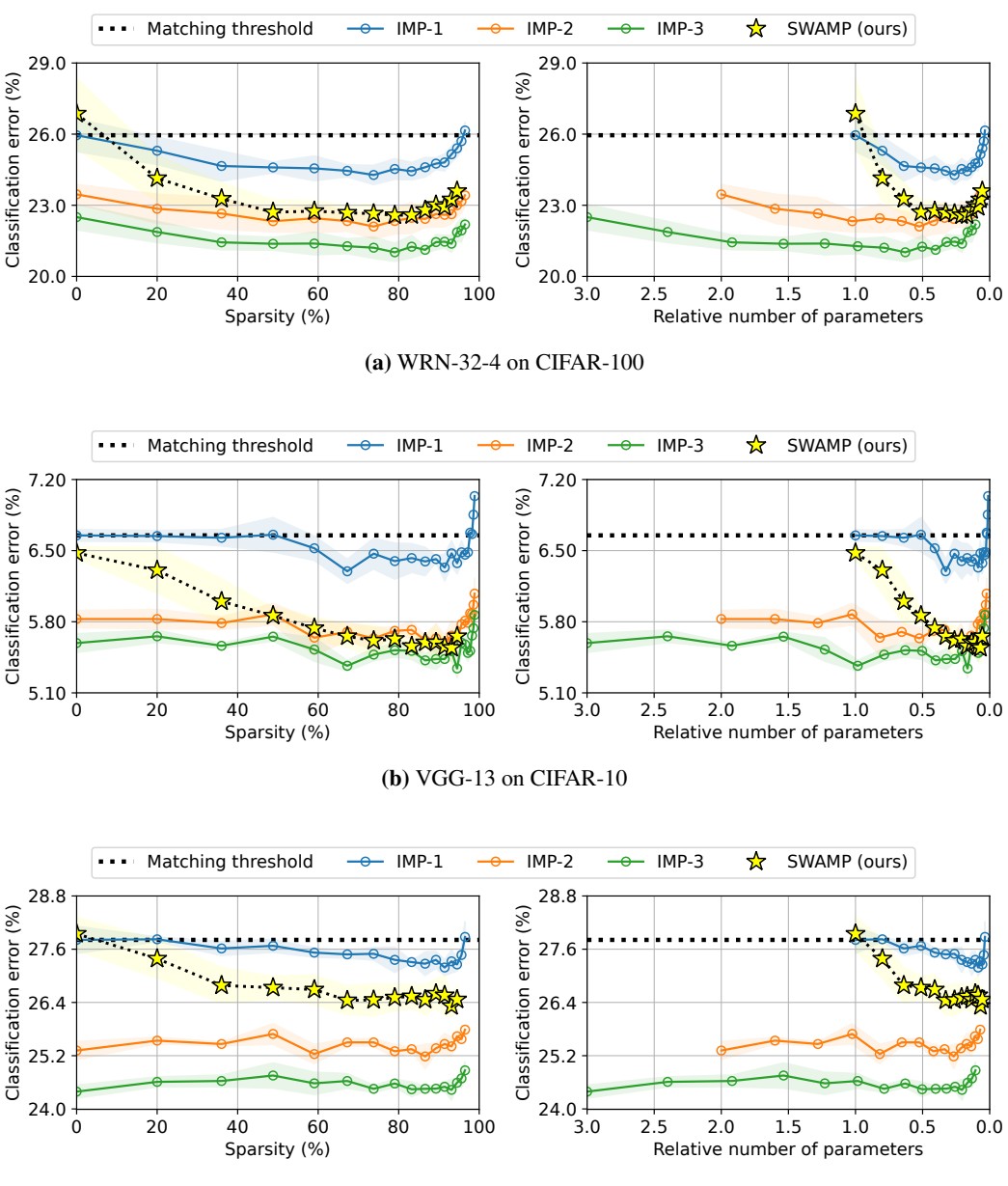

**(a)** WRN-32-4 on CIFAR-100

**(b)** VGG-13 on CIFAR-10

**(c)** VGG-16 on CIFAR-100

**Figure 7:** Supplementary plots for Figure 1. Classification error as a function of the sparsity (left) and the relative number of parameters (right). Again, SWAMP demonstrates *matching* performance even at extremely sparse levels, unlike IMP.

**Table 15:** Negative log-likelihood on CIFAR-10. SWAMP outperforms all the baselines across varying sparsities, including one-shot and dynamic pruning approaches. Reported values are averaged over three random seeds, and the best and second-best results are boldfaced and underlined, respectively.

|  | | Sparsity | | | |
|---|---|---|---|---|---|
|  | Method | 50% | 75% | 90% | 95% |
| CIFAR-10 (WRN-28-2) | SNIP | $0.227 \pm 0.012$ | $0.242 \pm 0.005$ | $0.282 \pm 0.003$ | $0.327 \pm 0.007$ |
| | SynFlow | $0.219 \pm 0.007$ | $0.223 \pm 0.004$ | $0.280 \pm 0.006$ | $0.316 \pm 0.003$ |
| | GraSP | $0.243 \pm 0.002$ | $0.281 \pm 0.009$ | $0.308 \pm 0.017$ | $0.338 \pm 0.004$ |
| | RigL | $0.206 \pm 0.010$ | $0.231 \pm 0.005$ | $0.261 \pm 0.009$ | $0.288 \pm 0.003$ |
| | DST | $0.189 \pm 0.009$ | $0.214 \pm 0.008$ | $0.241 \pm 0.010$ | $0.251 \pm 0.008$ |
| | IMP | $0.198 \pm 0.001$ | $0.198 \pm 0.002$ | $0.197 \pm 0.004$ | $\underline{0.210} \pm 0.003$ |
| | IMP + SAM | $\underline{0.182} \pm 0.009$ | $\underline{0.180} \pm 0.012$ | $\underline{0.187} \pm 0.011$ | $0.237 \pm 0.053$ |
| | **SWAMP (ours)** | $\mathbf{0.158 \pm 0.002}$ | $\mathbf{0.156 \pm 0.002}$ | $\mathbf{0.162 \pm 0.002}$ | $\mathbf{0.177 \pm 0.002}$ |
| CIFAR-10 (VGG-13) | SNIP | $0.234 \pm 0.003$ | $0.244 \pm 0.008$ | $0.269 \pm 0.003$ | $0.291 \pm 0.008$ |
| | SynFlow | $0.233 \pm 0.006$ | $0.224 \pm 0.008$ | $0.230 \pm 0.007$ | $0.261 \pm 0.002$ |
| | GraSP | $0.252 \pm 0.002$ | $0.257 \pm 0.006$ | $0.271 \pm 0.006$ | $0.282 \pm 0.001$ |
| | RigL | $0.216 \pm 0.002$ | $0.230 \pm 0.006$ | $0.246 \pm 0.007$ | $0.270 \pm 0.002$ |
| | DST | $0.210 \pm 0.002$ | $0.208 \pm 0.004$ | $0.213 \pm 0.002$ | $0.220 \pm 0.002$ |
| | IMP | $0.220 \pm 0.006$ | $0.214 \pm 0.004$ | $0.211 \pm 0.002$ | $0.210 \pm 0.004$ |
| | IMP + SAM | $\underline{0.195} \pm 0.001$ | $\underline{0.191} \pm 0.003$ | $\underline{0.190} \pm 0.003$ | $\underline{0.192} \pm 0.006$ |
| | **SWAMP (ours)** | $\mathbf{0.185 \pm 0.002}$ | $\mathbf{0.179 \pm 0.002}$ | $\mathbf{0.177 \pm 0.004}$ | $\mathbf{0.178 \pm 0.001}$ |

**Table 16:** Negative log-likelihood on CIFAR-100. SWAMP outperforms all the baselines across varying sparsities, including one-shot and dynamic pruning approaches. Reported values are averaged over three random seeds, and the best and second-best results are boldfaced and underlined, respectively.

|  | | Sparsity | | | |
|---|---|---|---|---|---|
|  | Method | 50% | 75% | 90% | 95% |
| CIFAR-100 (WRN-32-4) | SNIP | $1.067 \pm 0.010$ | $1.064 \pm 0.022$ | $1.121 \pm 0.036$ | $1.17 \pm 0.007$ |
| | SynFlow | $0.964 \pm 0.006$ | $1.000 \pm 0.019$ | $0.994 \pm 0.009$ | $1.009 \pm 0.041$ |
| | GraSP | $1.112 \pm 0.043$ | $1.198 \pm 0.028$ | $1.235 \pm 0.014$ | $1.252 \pm 0.036$ |
| | RigL | $1.005 \pm 0.012$ | $1.047 \pm 0.016$ | $1.105 \pm 0.029$ | $1.125 \pm 0.026$ |
| | DST | $1.012 \pm 0.036$ | $1.029 \pm 0.028$ | $1.235 \pm 0.014$ | $1.252 \pm 0.036$ |
| | IMP | $0.974 \pm 0.014$ | $0.973 \pm 0.007$ | $1.001 \pm 0.014$ | $1.010 \pm 0.011$ |
| | IMP + SAM | $\underline{0.924} \pm 0.031$ | $\underline{0.918} \pm 0.028$ | $\underline{0.934} \pm 0.024$ | $\underline{0.953} \pm 0.023$ |
| | **SWAMP (ours)** | $\mathbf{0.847 \pm 0.020}$ | $\mathbf{0.851 \pm 0.016}$ | $\mathbf{0.868 \pm 0.016}$ | $\mathbf{0.893 \pm 0.020}$ |
| CIFAR-100 (VGG-16) | SNIP | $1.135 \pm 0.015$ | $1.150 \pm 0.008$ | $1.216 \pm 0.012$ | $1.259 \pm 0.025$ |
| | SynFlow | $1.123 \pm 0.005$ | $1.115 \pm 0.006$ | $1.092 \pm 0.010$ | $1.203 \pm 0.004$ |
| | GraSP | $1.219 \pm 0.003$ | $1.242 \pm 0.013$ | $1.311 \pm 0.016$ | $1.351 \pm 0.005$ |
| | RigL | $1.121 \pm 0.007$ | $1.142 \pm 0.007$ | $1.120 \pm 0.016$ | $1.238 \pm 0.004$ |
| | DST | $1.110 \pm 0.007$ | $1.121 \pm 0.004$ | $1.138 \pm 0.004$ | $1.146 \pm 0.014$ |
| | IMP | $1.105 \pm 0.006$ | $1.096 \pm 0.009$ | $1.090 \pm 0.004$ | $1.096 \pm 0.002$ |
| | IMP + SAM | $\underline{1.075} \pm 0.014$ | $\underline{1.067} \pm 0.010$ | $\underline{1.071} \pm 0.004$ | $\underline{1.068} \pm 0.007$ |
| | **SWAMP (ours)** | $\mathbf{1.023 \pm 0.016}$ | $\mathbf{1.029 \pm 0.015}$ | $\mathbf{1.033 \pm 0.016}$ | $\mathbf{1.029 \pm 0.016}$ |

**Table 17:** Trace of Hessian Tr(**H**) evaluated across training data. In most cases, SWAMP with multiple particles exhibits smaller trace value, i.e., finds flatter minima, compared to others. Reported values are averaged over three random seeds, and the best and second-best results are boldfaced and underlined, respectively.

|  | Training | Sparsity | | | |
|---|---|---|---|---|---|
|  |  | 20% | 50% | 75% | 90% |
| CIFAR-100 (WRN-32-4) | SGD | 16394.8 ± 3262.1 | 20335.2 ± 4875.3 | 29337.7 ±10887.7 | 25504.9 ± 4370.0 |
|  | SWAMP ($N = 1$) | 2586.9 ± 10.7 | 2669.4 ± 21.9 | 3079.8 ± 150.8 | 3159.3 ± 147.3 |
|  | SWAMP ($N = 4$) | **2457.8** ± 161.8 | **2556.1** ± 86.9 | **2896.7** ± 201.0 | **2968.5** ± 79.4 |
| CIFAR-100 (VGG-16) | SGD | 4776.8 ± 80.5 | 4842.5 ±108.0 | 5033.1 ±336.3 | 5141.7 ± 20.5 |
|  | SWAMP ($N = 1$) | 2072.4 ± 34.5 | **2189.2** ± 73.9 | 2472.8 ± 93.1 | 2570.1 ± 88.8 |
|  | SWAMP ($N = 4$) | **1995.9** ± 39.6 | 2201.1 ± 85.5 | **2341.6** ±109.6 | **2423.8** ± 48.6 |

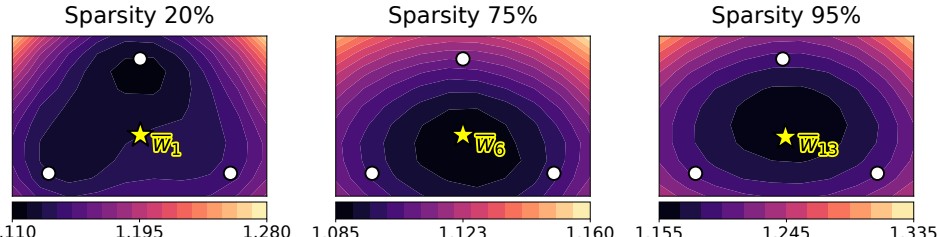

**Figure 8:** Visualization of loss surfaces as a function of neural network weights in a two-dimensional subspace, spanned by three particles (marked as white circles). The averaged weight $w_c$ (marked by a yellow star) is observed not to be positioned in the flat region of the surface during the earlier stages of IMP (left; Sparsity 20%). However, as the sparsity increased, the weight averaging technique effectively captures the flat region of the surface. The results are presented for WRN-32-4 on the test split of CIFAR-100.

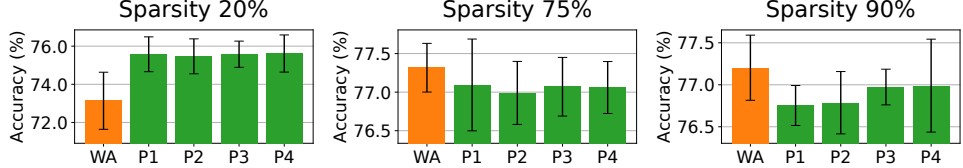

**Figure 9:** Bar plots depicting the accuracy of individual particles involved in the averaging process of the SWAMP algorithm. While the averaged weight (denoted as WA) may not outperform individual particles (denoted as P1-P4) in the early stages of IMP (left; Sparsity 20%), it achieves high performance at high sparsity levels. The results are presented for WRN-32-4 on the test split of CIFAR-100.

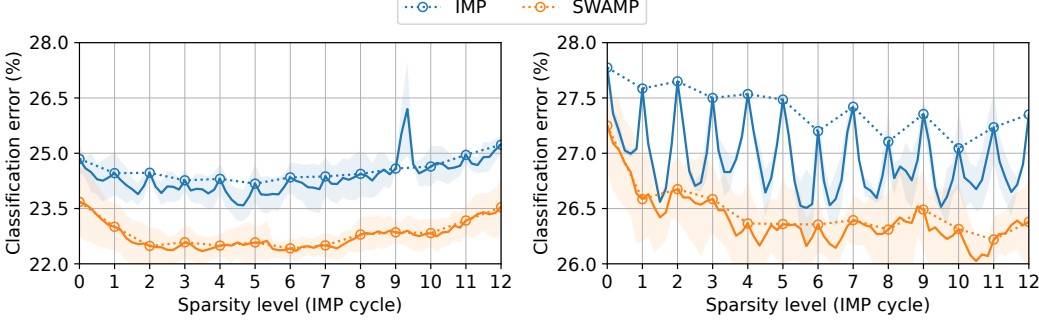

**Figure 10:** Linear connectivity between sparse solutions having different sparsity along with the IMP cycle. The results are presented for WRN-32-4 (left) and VGG-16 on the test split of CIFAR-100.

