# OpenReview forum: "Sparse Weight Averaging with Multiple Particles for Iterative Magnitude Pruning"
_ICLR.cc/2024/Conference — ICLR 2024 poster_

### Official Review · Reviewer_Eb3S · 2023-10-31

**Soundness:** 2 fair
**Presentation:** 3 good
**Contribution:** 2 fair
**Rating:** 6
**Confidence:** 4

**Summary:**

The paper presents SWAMP, a novel method using the average of multiple particles' stochastic weight averaging (SWA) to achieve improved model performance. The authors have tested SWAMP’s effectiveness across various tasks, including vision models (CNNs) and language models (RoBerTa finetuning), providing a comprehensive evaluation.

**Strengths:**

1. The paper is well-written, and easy-to-follow.
2. The authors have conducted extensive experiments, covering vision tasks and language model fine-tuning. Additional studies such as mask analysis and efficient implementation SWAMP+ are also provided.

**Weaknesses:**

While the authors address the computational efficiency of SWAMP+ in section 4.3, stating that it can utilize a single particle for the first few iterations, this claim seem to work because the networks are already very sparse. My concern lies in the computational cost of SWAMP+ at lower sparsity levels. Furthermore, did the authors test SWAMP+ on ImageNet?

See also my questions.

I am willing to adjust my rating if my questions are addressed.

**Questions:**

1. The results in Figure 4 show that interpolated weights yield even lower errors compared to IMP weights. Could the authors provide a detailed explanation or hypothesis as to why this is the case?
2. Appendix B states, "The learning rate for this phase (SWA phase) is set to a constant value of 0.05." Does this imply that the minimum learning rate is set at 0.05 for SWAMP, and for other baselines such as IMP?
3. Do IMP and SWAMP use the same epoch T_0 to rewind weights?

---

> ### Author Response · Authors · 2023-11-14
>
> __[W1] *(Re. SWAMP+ in low sparsity regime):*__ Thank you for pointing out the effectiveness of SWAMP+ in low sparsity regimes. We agree with the reviewer that the effectiveness of SWAMP+ fades under low sparsity regimes. That said, IMP is a SOTA algorithm for extremely high sparse networks (sparsity over 90%). We thus suggest two alternatives instead of SWAMP+ for low sparsity regimes. Firstly, dynamic sparse methods (e.g., RigL) can also benefit from SWAMP as shown in Table 9 which exhibits significantly lower training costs than IMP. Secondly, leveraging the robustness of SWAMP (refer to Table 7), one can employ higher pruning ratio which can still boost the training speed even in low sparsity regimes. We believe the general response could further resolve your concerns regarding the training cost of the algorithm. Additionally, as per your great suggestion, we hereby provide the results of IMP+ on ImageNet in Table R.10. Please note that the results in Table R.10 differ from Table 3, as the experiment was conducted earlier with a different set of hyperparameters.
>
> __[Q1] *(Re. superior performance of interpolated networks in Fig. 4):*__ We appreciate the reviewer for highlighting empirical analysis in Figure 4. Following [Paul et al., 2023], Figure 4 illustrates that there exists a linear path on the loss landscape connecting two successive IMP/SWAMP solutions. In case of both IMP/SWAMP, as the reviewer pointed out, the interpolated weights exhibit lower loss compared to obtained solutions which is also corroborated by several prior works [Garipov et al., 2018; Wortsman et al., 2021; Nam et al., 2022; Paul et al., 2023]. This finding suggests that one can obtain a better solution interpolating successive solutions which is the main motivation for Lottery Pools [Yin et al., 2023]. Further, while we only highlight the existence of linear connectors in the paper, we here provide additional experimental results in Tables R.11 and R.12 suggesting that SWAMP can also benefit from such a phenomenon, i.e., combining SWAMP and Lottery Pools leads to further performance enhancement.
>
> __[Q2] *(Clarification: learning rate setting):*__ Thank you for the attentive review. In our IMP experiments, we explored two learning rate schedules: cosine annealing (initiating from 0.1 and decaying to 0.001) and step learning rate (adopted from SWA; dropping to a constant value of 0.05). The cosine annealing schedule turned out to be the best for IMP, while SWAMP exhibited better performance with step lr schedule. In short, we employed the cosine annealing schedule for IMP and the step lr schedule for SWAMP.
>
> __[Q3] *(Clarification: rewinding epochs):*__ Yes, indeed. We set the same matching epoch for both IMP and SWAMP.
>
> \
> __Table R.10.__ Results of SWAMP+ on ImageNet.
> | Method | 86.4% | 89.1% | 91.2% | 92.9% | 94.3% |
> | :-     | :-    | :-    | :-    | :-    | :-    |
> | IMP    | 76.15 | 76.02 | 75.48 | 75.16 | 74.43 |
> | SWAMP+ | 76.55 | 76.33 | 76.06 | 75.45 | 74.84 |
>
> \
> __Table R.11.__ Further comparison between (a) IMP, (b) SWAMP, (c) Lottery Pools, and (d) SWAMP + Lottery Pools algorithm on CIFAR-10.
> |     | Sparsity 50% | Sparsity 75% | Sparsity 90% | Sparsity 95% |
> | :-  | :-:          | :-:          | :-:          | :-:          |
> | (a) | 93.97 ± 0.16 | 94.02 ± 0.23 | 93.90 ± 0.15 | 93.58 ± 0.09 |
> | (b) | 94.74 ± 0.04 | 94.88 ± 0.09 | 94.73 ± 0.10 | 94.23 ± 0.11 |
> | (c) | 94.39 ± 0.16 | 94.28 ± 0.14 | 94.16 ± 0.11 | 93.43 ± 0.23 |
> | (d) | 94.65 ± 0.14 | 94.70 ± 0.25 | 94.52 ± 0.25 | 94.31 ± 0.28 |
>
> \
> __Table R.12.__ Further comparison between (a) IMP, (b) SWAMP, (c) Lottery Pools, and (d) SWAMP + Lottery Pools algorithm on CIFAR-100.
> |     | Sparsity 50% | Sparsity 75% | Sparsity 90% | Sparsity 95% |
> | :-  | :-:          | :-:          | :-:          | :-:          |
> | (a) | 75.40 ± 0.23 | 75.72 ± 0.41 | 75.24 ± 0.25 | 74.60 ± 0.37 |
> | (b) | 77.29 ± 0.53 | 77.35 ± 0.39 | 77.14 ± 0.33 | 76.48 ± 0.73 |
> | (c) | 76.31 ± 0.51 | 76.17 ± 1.03 | 75.84 ± 0.67 | 75.14 ± 0.49 |
> | (d) | 77.52 ± 0.46 | 77.45 ± 0.53 | 77.36 ± 0.61 | 77.10 ± 0.54 |
>
>
> \
> __References__
> - Garipov et al., “Loss Surfaces, Mode Connectivity, and Fast Ensembling of DNNs.” NeurIPS 2018.
> - Wortsman et al., “Learning Neural Network Subspaces.” ICML 2021.
> - Nam et al., “Improving Ensemble Distillation With Weight Averaging and Diversifying Perturbation.” ICML 2022.
> - Paul et al., “Unmasking the Lottery Ticket Hypothesis: What's Encoded in a Winning Ticket's Mask?” ICLR 2023.
> - Yin et al., “Lottery Pools: Winning More by Interpolating Ticket without Increasing Training or Inference Cost.” AAAI 2023.

---

> > ### Comment · Reviewer_Eb3S · 2023-11-20
> > **Thank you for your response**
> >
> > I thank the authors for their responses. I have the following additional questions.
> >
> > ### 1. W1 and training costs.
> >
> > **No additional training costs of parallelization?** While I understand your perspective, I maintain a different view regarding the impact of parallelization. Parallelization transforms time costs into the costs of using multiple GPUs, which should be accounted for.
> >
> > **SWAMP+.** Regarding SWAMP+, the addition of results in Table R.10 is appreciated. Could the authors possibly provide a comparison of FLOPs between IMP and SWAMP+ in this table?
> >
> > ### 2. Q2 and unfair comparison
> >
> > Thank you for your response to Q2. I appreciate the information shared. However, to address concerns about potentially unfair comparisons, could the authors provide the results of IMP with the step learning rate, such as in CIFAR-10/100?
> >
> > ### 3. Q1 and Q3
> >
> > Thank you for your detailed clarification.

---

> > > ### Author Response · Authors · 2023-11-21
> > >
> > > __[Q1] *(Memory cost of SWAMP):*__
> > > Thank you for the additional question. We certainly agree with your concern regarding the increased memory demand for SWAMP. That said, we want to emphasize two key points: (i) SWAMP maintains practical training speed while delivering significant improvements in performance, and (ii) we believe that additional memory cost (in small to medium scale experiments) is not a notable drawback, as large scale experiments typically demand a distributed training setting where models are already distributed across multiple copies. As to why distributed learning does not incur additional memory cost, we further discuss in the following section.
> > >
> > > __[Q2] *(Update FLOPs in Table R.10):*__
> > > As mentioned in the earlier response, the checkpoints for the models in Table R.10 are unavailable (as they are outcomes from the initial phases of the research and were unfortunately deleted to clear disk space). Instead, we offer the training FLOPs of models provided in Table R.2 without incorporating the SWAMP+ approach.
> > >
> > > __Table R.13.__ Cumulative GFLOPs of IMP and SWAMP for ImageNet (it supplements Table R.2).
> > > | Method | 45.9% | 68.8% | 80.3% | 86.0% |
> > > | :-     | :-    | :-    | :-    | :-    |
> > > | IMP    | 6.40  | 7.73  | 8.52  | 9.01  |
> > > | SWAMP  | 6.40  | 7.74  | 8.54  | 9.03  |
> > >
> > > Because we carried out our ImageNet experiments in a distributed training environment with eight machines, SWAMP demands virtually no additional FLOPs compared to IMP (and SWAMP+ would also be at the same level as IMP and SWAMP). For clearer understanding, we provide more details on distributed training:
> > > 1. In distributed training, considering a mini-batch size of 2048 for one training step, each machine calculates gradients for 256 (=2048/8) examples. In the case of IMP, these gradients are averaged across machines using the all-reduce mean operation, ensuring consistent updates to the model copies on each machine.
> > > 2. In this scenario, we discovered that conceptualizing each machine as overseeing a single particle allowed us to develop a parallelization strategy for the SWAMP algorithm. The approach to treating multiple model copies across machines as multiple particles is straightforward—just skip the step of gradient synchronization through the all-reduce mean operation. Although each particle processes fewer samples, the weight-averaging applied at the end of each training epoch effectively compensates for this, thereby enhancing  the model performance.
> > > 3. As a result, SWAMP, with the applied parallelization strategy, undergoes the following cost adjustments compared to IMP: (-) The need for the cost of gradient synchronization through the all-reduce operation at each training step is eliminated. (+) The cost of weight averaging for all particles is now incurred at every training epoch.
> > > 4. Both these expenses constitute a relatively minor fraction of the entire training process. Consequently, in a distributed training setting, the parallelized SWAMP does not entail extra training costs when contrasted with IMP. Hence, in a large-scale experimental configuration built on the assumption of a distributed training environment, we can ensure the scalability of SWAMP.
> > >
> > > We hope that we have adequately addressed your concerns.
> > >
> > > __[Q3] *(IMP with step-wise learning rate schedule):*__
> > > Thank you for the constructive feedback! We further present IMP with a step-wise LR schedule in Tables R.14 and R.15. We can find that IMP exhibits better performance with the original cosine LR schedule rather than step LR schedule. We employed a multi-step learning rate schedule decayed by a factor of 0.1 at 75 and 113 epochs (with a total of 150 epochs).
> > >
> > > __Table R.14.__ Further comparison between (a) IMP, (b) IMP with step-wise LR, (c) SWAMP on CIFAR-10.
> > > |     | Sparsity 50% | Sparsity 75% | Sparsity 90% | Sparsity 95% |
> > > | :-  | :-:          | :-:          | :-:          | :-:          |
> > > | (a) | 93.97 ± 0.16 | 94.02 ± 0.23 | 93.90 ± 0.15 | 93.58 ± 0.09 |
> > > | (b) | 93.86 ± 0.16 | 93.93 ± 0.12 | 93.74 ± 0.10 | 93.46 ± 0.08 |
> > > | (c) | 94.74 ± 0.04 | 94.88 ± 0.09 | 94.73 ± 0.10 | 94.23 ± 0.11 |
> > >
> > > __Table R.15.__ Further comparison between (a) IMP, (b) IMP with step-wise LR, (c) SWAMP on CIFAR-100.
> > > |     | Sparsity 50% | Sparsity 75% | Sparsity 90% | Sparsity 95% |
> > > | :-  | :-:          | :-:          | :-:          | :-:          |
> > > | (a) | 75.40 ± 0.23 | 75.72 ± 0.41 | 75.24 ± 0.25 | 74.60 ± 0.37 |
> > > | (b) | 74.72 ± 0.64 | 74.91 ± 0.17 | 74.90 ± 0.38 | 74.54 ± 0.09 |
> > > | (c) | 77.29 ± 0.53 | 77.35 ± 0.39 | 77.14 ± 0.33 | 76.48 ± 0.73 |

---

> > > > ### Comment · Reviewer_Eb3S · 2023-11-21
> > > > **Thank you for your reply**
> > > >
> > > > Thank you for your clarification. Most of my concerns are addressed and I have adjusted my rating from 5 to 6.

---

### Official Review · Reviewer_1f6B · 2023-10-31

**Soundness:** 3 good
**Presentation:** 3 good
**Contribution:** 2 fair
**Rating:** 6
**Confidence:** 4

**Summary:**

This paper extends the Iterative Magnitude Pruning (IMP) technique by proposing an approach called SWAMP that trains multiple sparse models called particles in each magnitude pruning iteration of IMP using Stochastic Weighted Averaging (SWA) optimization. The particles in each pruning iteration exhibit the same matching ticket and their diversity is achieved through different batch orders. The trained particle masks are combined in the weighted average fashion to get the single mask of a given pruning iteration. This process of training multiple particles followed by the weighted average of their mask is repeated until desired sparsity or pruning iteration is achieved. The experimentation is conducted on multiple datasets along with different architectures to showcase the effectiveness of the proposed SWAMP model.

**Strengths:**

* The authors have done a great job in terms of summarizing their contributions compared to the IMP technique in Section 3.2.
* The paper is very easy to read and the proposed contribution can be easily understood through Algorithm 1.
* Extensive experimentation is conducted on multiple tasks (vision and language), multiple datasets, and multiple architectures.
* A very comprehensive ablation study is conducted to showcase the effectiveness of the proposed components in the paper. For example, Table  5 clearly shows the importance of the SWA optimization along with the weighted average mechanism of the particles to enhance the performance.

**Weaknesses:**

* In terms of methodology, the proposed technique provides an empirically guided straightforward extension over the IMP technique. The proposed SWAMP therefore has a trivial contribution and therefore lacks novelty.
* In terms of experimentation, the performance gain over other techniques seems to be marginal and  reduces the significance of their proposed methodology.
* In Figure 3, for relatively lower sparsity (e.g., sparsity of 20%), the proposed Weighted Average (WA) technique seems to underperform the individual particle performance. Does this mean, the proposed technique  harm the performance on the lower sparsity? The authors may need to provide more extensive justification to explain this phenomenon.

**Questions:**

In Figure 3, why does the proposed technique have a lower performance compared to individual particles in the lower network sparsity?

---

> ### Author Response · Authors · 2023-11-14
>
> __[W1] *(Re. lack of novelty):*__ Our primary contribution extends beyond the development of an algorithm; it involves empirically demonstrating that sparse networks trained with varying SGD noise can benefit from weight-averaging, especially as they become sparser – a finding that is not immediately obvious and has been acknowledged by Reviewer-tk5N. Moreover, as Reviewer-MGna highlighted, our work is inspired by the theoretical framework of [Paul et. al, 2023] which enabled us to empirically validate whether SWAMP retains the key property of IMP; we kindly refer the reviewer to Section 3.3.
>
> __[W2] *(Marginal improvement of the algorithm):*__ Please refer to General Response B.
>
> __[W3, Q1] *(Analysis under lower sparsity regime):*__ Thank you for highlighting the failure of sparse weight averaging under low sparsity regimes. We can see that weight averaging fails at the earlier stages of IMP due to the highly non-convex nature of the landscape. However, as sparsity increases, particles tend to locate in the same wide basin which enables weight-averaging. Such a finding is in line with [Frankle et al., 2020; Paul et al., 2023] demonstrating the ease of finding a low-loss curve with a smaller network compared to a larger one, i.e., a sparse network tends to be more stable. Additionally, it further supports that our algorithm benefits more with sparser networks.
>
> \
> __References__
> - Frankle et al., “Linear mode connectivity and the lottery ticket hypothesis.” ICML 2020.
> - Paul et al., “Unmasking the Lottery Ticket Hypothesis: What's Encoded in a Winning Ticket's Mask?” ICLR 2023.

---

> > ### Author Response · Authors · 2023-11-21
> > **Reminder**
> >
> > Dear reviewer-1f6B,
> >
> > We sincerely appreciate your time and the constructive feedback. With the Author/Reviewer discussion deadline approaching, we would be grateful if you let us know whether our responses have addressed your concerns. This will greatly assist us in enhancing our work. For any further clarification, we are more than happy to answer.
> >
> > Warm regards, \
> > Authors

---

> > > ### Comment · Reviewer_1f6B · 2023-11-22
> > > **Response to the Rebuttal**
> > >
> > > I would like to thank the authors for the clarification.  While I maintain my view that the proposed technique has limited novelty, I commend the authors for effectively justifying the empirical results during the rebuttal phase. Therefore, I am inclined to increase the score from 5 to 6.

---

### Official Review · Reviewer_YhLg · 2023-11-01

**Soundness:** 3 good
**Presentation:** 3 good
**Contribution:** 2 fair
**Rating:** 6
**Confidence:** 3

**Summary:**

This paper proposes a modification to the Iterative Magnitude Pruning algorithm, SWAMP. The basis of this algorithm is the empirical evidence that different models trained from the same matching tickets can be weight averaged without encountering a loss barrier post certain sparsity levels. SWAMP obtains marginal accuracy improvements with respect to the baselines used.

**Strengths:**

S1. The manuscript is well written

S2. The method is empirically sound and arguments are well made.

S3. Extensive empirical is provided to justify the merit in this approach.

**Weaknesses:**

W1. The authors have not empirically justified their choice of using Stochastic Weight Averaging (SWA) as opposed to SGD in the manuscript. It would be important to understand the impact of SWA on the proposed approach by demonstrating two things.
1. How does IMP perform when it uses SWA as opposed to SGD.
2. How does SWAMP perform when it uses SGD as opposed to SWA.

W2. Multiple instances of imprecise statements. For example, "As illustrated in Figure 1, our algorithm achieves superior performance, which is on par with that of an ensemble consisting of two sparse networks." It is not clear with respect to what are the authors claiming superior performance? Because in Figure 1, IMP-3 outperforms SWAMP in terms of accuracy.

**Questions:**

Q1. I would like to understand why is it that the authors choose to average the weights in SWAMP? As demonstrated in Figure 1, there might be individual IMP runs that outperform SWAMP. Why not take the best of multiple pruned weights?

---

> ### Author Response · Authors · 2023-11-14
>
> __[W1] *(Ablation study w.r.t. SWA & multiple particles):*__ Thank you for pointing out the issue. We indeed provided an ablation study regarding SWA in Table 5. In detail, the second and the third row each correspond to SWAMP without SWA and IMP with SWA, respectively. We can readily find that each component, SWA and multi-particles, contributes similarly to the performance gain, and the optimal performance is attained when used in combination.
>
> __[W2] *(Clarity: performance comparison to ensembles):*__ We greatly appreciate the reviewer for pointing out unclear points. In Figure 1, we show that SWAMP matches the performance of an ensemble of two sparse networks but not surpassing that of an ensemble of three networks, which is still a significant performance gain considering impressive performance of Deep Ensembles [Lakshminarayanan et al., 2017; Fort et al., 2019]. Further, in the right side of Figure 1, we demonstrate how SWAMP offers advantages over ensembles in terms of memory costs.
>
> __[Q1] *(Re. best seed performance):*__   As per your insightful suggestion, we provide the performance of an IMP ensemble, wherein each member is chosen as the best-performing model from multiple random seeds. Tables R.8 and R.9 below verify that exploring more random seeds does not result in additional improvements.
>
> \
> __Table R.8.__ Further comparison between (a) IMP, (b) IMP selected from best seed, and (c) SWAMP  algorithm on CIFAR-10.
> |     | Sparsity 50% | Sparsity 75% | Sparsity 90% | Sparsity 95% |
> | :-  | :-:          | :-:          | :-:          | :-:          |
> | (a) | 93.97 ± 0.16 | 94.02 ± 0.23 | 93.90 ± 0.15 | 93.58 ± 0.09 |
> | (b) | 93.91 ± 0.23 | 94.14 ± 0.13 | 93.81 ± 0.14 | 93.72 ± 0.27 |
> | (c) | 94.74 ± 0.04 | 94.88 ± 0.09 | 94.73 ± 0.10 | 94.23 ± 0.11 |
>
> __Table R.9.__ Further comparison between (a) IMP, (b) IMP selected from best seed, and (c) SWAMP  algorithm on CIFAR-100.
> |     | Sparsity 50% | Sparsity 75% | Sparsity 90% | Sparsity 95% |
> | :-  | :-:          | :-:          | :-:          | :-:          |
> | (a) | 75.40 ± 0.23 | 75.72 ± 0.41 | 75.24 ± 0.25 | 74.60 ± 0.37 |
> | (b) | 75.79 ± 0.53 | 75.05 ± 0.39 | 74.92 ± 0.58 | 74.35 ± 0.43 |
> | (c) | 77.29 ± 0.53 | 77.35 ± 0.39 | 77.14 ± 0.33 | 76.48 ± 0.73 |
>
>
> \
> __References__
> - Lakshminarayanan et al., “Simple and Scalable Predictive Uncertainty Estimation using Deep Ensembles.” NeurIPS 2017.
> - Fort et al., “Deep Ensembles: A Loss Landscape Perspective.” arXiv preprint, 2019.

---

> > ### Author Response · Authors · 2023-11-21
> > **Reminder**
> >
> > Dear reviewer-YhLg,
> >
> > We sincerely appreciate your time and the constructive feedback. With the Author/Reviewer discussion deadline approaching, we would be grateful if you let us know whether our responses have addressed your concerns. This will greatly assist us in enhancing our work. For any further clarification, we are more than happy to answer.
> >
> > Warm regards, \
> > Authors

---

> ### Comment · Reviewer_YhLg · 2023-11-22
> **Thank you for your rebuttal**
>
> Thank you for your rebuttal. My concerns have been addressed. I request the authors to fix imprecise statements in the manuscript such as that mentioned W2. I also request the authors to provide recent and relevant citations and a more precise replacement for their statement on IMP being state-of-the-art in their abstract (eg. IMP is SoTA in unstructured pruning at high-sparsity). Because this is not true in the structured sparsity regime [a].
>
> I will maintain my score.
>
> [a] Neural Pruning via Growing Regularization, Wang et al. ICLR 2021

---

### Official Review · Reviewer_tk5N · 2023-11-01

**Soundness:** 4 excellent
**Presentation:** 4 excellent
**Contribution:** 2 fair
**Rating:** 8
**Confidence:** 4

**Summary:**

The authors propose weight averaging of sparse models trained from a checkpoint of a single model, in many ways "model soups" for Iterative Maginitude Pruning (IMP). The authors motivate the method for IMP as as model soups in the dense context are, with the loss landscape perspective: we know that LTs lie within the same loss basin, and might expect that weight averaging would find a better generalizing solution. Experiments demonstrate that the approach identifies solutions within a flatter region of the loss basin, and improved generalization over IMP and many other sparse training methods for CIFAR-10/100 and ImageNet models.

**Strengths:**

* The paper is overall well-written, with a good organization, clear writing for the most part, and a clear methodology.
* The experimental analysis is appropriate, using reasonable datasets and models (except VGG), and demonstrates clear knowledge of the sparse training literature appropriate to the methodology.
* The paper has a clear and well defined motivation: the method is motivated as cheaper than ensembles, much along the same lines of the model soups paper and how it is motivated for dense training.
* The loss landscape analysis also originally used in model soups is clearly applicable to the sparse domain, especially since much of the linear-mode connectivity methodology comes from the sparse literature to begin with.
* Hessian Trace analysis also provides some signal that the loss-landscape motivation for weight averaging holds in the sparse realm.

**Weaknesses:**

* The method comes down to applying the model soup paper to sparse training/IMP. I believe there is sufficient novelty in applying a method only shown on dense training and not necessarily repeatable in the sparse training context, never mind the extensive analysis shown by the authors in this work. Saying that, it's also not the most novel research direction out there compared to many papers.
* As presented in the main body of the paper, SWAMP is *much* more expensive than most of the compared sparse training methods in e.g. Table 2 at *training time*. This is because IMP with weight rewinding is extraordinarily expensive in practice. However, the authors do demonstrate that the SWAMP methodology applies to other much more efficient sparse training methods in the appendix, notably RiGL, a state-of-the-art sparse training method, and one that is reasonably efficient. I believe the authors should focus their method as being widely applicable to sparse training methods in the main body of the paper, rather than focusing on IMP however - this is especially important given the motivation that SWAMP is better than training an ensemble (which is in fact likely cheaper than SWAMP when using more practical sparse training methods than IMP!).
* While CIFAR-10/100 results are relatively strong, the ImageNet results (Table 3) are relatively quite weak and not as obviously significant.

**Questions:**

* While the paper is motivated by comparing the generalization of a SWAMP to an ensemble of two IMP solutions, what is the comparison in generalization when using other sparse training methods, e.g. RiGL, given that these other methods often generalize better than IMP?
* Is there any reason to believe SWAMP is not a general method that applies to any sparse training method? If so what? If not, why focus on IMP?

---

> ### Author Response · Authors · 2023-11-14
>
> __[W1] *(Extension to different hyperparameters):*__ We thank the reviewer for recognizing the novelty of our work, where our main objective is to empirically validate sparse weight averaging technique and whether it preserves the characteristics of IMP rooted in the theoretical framework of [Paul et al., 2023]. As the reviewer pointed out the work of model soups [Wortsman et al., 2022], we hereby provide extended results of SWAMP with multiple particles obtained from various hyperparameters settings (weight decaying rate, learning rate) beyond random seeds in Table R.5. Our findings indicate that weight averaging with particles perturbed by means other than random seeds do not markedly improve performance. We acknowledge that further optimization of hyperparameters, given enough time, could potentially boost SWAMP's performance making it an interesting direction for future research. That said, one should also consider whether the improvements of SWAMP variants justify the additional computational cost they demand.
>
> __[W2, Q2] *(Heavy training cost & extension to other pruning methods):*__ As per your constructive feedback, we will revise the final manuscripts to include the SWAMP-extended dynamic pruning results in the main body of the paper, not the appendix. As shown in Table 9, SWAMP is indeed applicable to other sparse training methods. During the rebuttal phase, as time permits, we aim to investigate whether dynamic pruning methods, aside from RigL, can also benefit from SWAMP. Moreover, we choose to focus on IMP in order to verify whether SWAMP preserves the theoretical characteristics outlined in [Paul et al., 2023] as discussed in Section 3.3. Also, we believe that General Response A could further resolve your concerns regarding the training cost of the algorithm.
>
> __[W3] *(Marginal gain in ImageNet experiments):*__ Please refer to General Response B.
>
> __[Q1] *(Re. ensemble of dynamic pruning methods):*__  As per your detailed feedback, we hereby provide additional results on the ensemble of RigL and DST solutions. Tables R.6 and R.7 provide clear evidence that SWAMP exhibits comparable generalization to test sets when compared to ensembles of other baseline methods.
>
> \
> __Table R.5.__ Further comparison on Cifar-10 with ResNet-20 between (a) SWAMP particles with varying weight decaying rates, and (b) SWAMP particles with varying learning rates. Reported values are averaged over three random seeds.
> |     | Sparsity 50% | Sparsity 75% | Sparsity 90% | Sparsity 95% |
> | :-  | :-:          | :-:          | :-:          | :-:          |
> | IMP | 91.57 ± 0.08 | 91.51 ± 0.08 | 90.44 ± 0.16 | 89.07 ± 0.33 |
> | SWAMP | 92.68 ± 0.04 | 92.59 ± 0.20 | 92.03 ± 0.05 | 90.62 ± 0.06 |
> | (a) | 91.33 ± 0.29 | 91.84 ± 0.22 | 91.41 ± 0.22 | 90.17 ± 0.11 |
> | (b) | 92.48 ± 0.19 | 92.49 ± 0.12 | 91.82 ± 0.17 | 90.52 ± 0.09 |
>
> __Table R.6.__ Additional accuracy results on the ensemble of two particles of (a) VGG13 with RigL, (b) VGG13 with DST on CIFAR-10. And (c) is the performance of a single SWAMP solution. Reported values are averaged over three random seeds.
> |     | Sparsity 50% | Sparsity 75% | Sparsity 90% | Sparsity 95% |
> | :-  | :-:          | :-:          | :-:          | :-:          |
> | (a) | 94.34 ± 0.13 | 94.12 ± 0.15 | 94.15 ± 0.22 | 93.33 ± 0.09 |
> | (b) | 94.54 ± 0.09 | 94.48 ± 0.19 | 94.29 ± 0.14 | 94.11 ± 0.02 |
> | (c) | 94.14 ± 0.08 | 94.39 ± 0.15 | 94.40 ± 0.16 | 94.34 ± 0.11 |
>
> __Table R.7.__ Additional accuracy results on the ensemble of two particles of (a) VGG16 with RigL, (b) VGG16 with DST on CIFAR-100. And (c) is the performance of a single SWAMP solution. Reported values are averaged over three random seeds.
> |     | Sparsity 50% | Sparsity 75% | Sparsity 90% | Sparsity 95% |
> | :-  | :-:          | :-:          | :-:          | :-:          |
> | (a) | 74.43 ± 0.21 | 73.66 ± 0.20 | 73.15 ± 0.15 | 71.91 ± 0.09 |
> | (b) | 75.06 ± 0.14 | 74.87 ± 0.21 | 74.46 ± 0.16 | 73.67 ± 0.16 |
> | (c) | 73.27 ± 0.26 | 73.54 ± 0.36 | 73.40 ± 0.33 | 73.53 ± 0.32 |
>
> \
> __References__
> - Wortsman et al., “Model soups: averaging weights of multiple fine-tuned models improves accuracy without increasing inference time.” ICML 2022.
> - Paul et al., “Unmasking the Lottery Ticket Hypothesis: What's Encoded in a Winning Ticket's Mask?” ICLR 2023.

---

> > ### Comment · Reviewer_tk5N · 2023-11-22
> >
> > I'd like to thank the authors for their rebuttal, and apologize for my late participation in the rebuttal period - this was due to exceptional circumstances.
> >
> > I believe the authors have addressed most of my questions/concerns, and hopefully in incorporating the feedback they did, this has also strengthened the paper significantly.

---

> ### Author Response · Authors · 2023-11-21
> **Reminder**
>
> Dear reviewer-tk5N,
>
> We sincerely appreciate your time and the constructive feedback. With the Author/Reviewer discussion deadline approaching, we would be grateful if you let us know whether our responses have addressed your concerns. This will greatly assist us in enhancing our work. For any further clarification, we are more than happy to answer.
>
> Warm regards, \
> Authors

---

### Official Review · Reviewer_MGna · 2023-11-02

**Soundness:** 3 good
**Presentation:** 3 good
**Contribution:** 2 fair
**Rating:** 6
**Confidence:** 4

**Summary:**

SWAMP (Sparse Weight Averaging with Multiple Particles) is a new pruning method that enhances the performance of sparse neural networks by averaging multiple models trained with different stochastic gradients but sharing an identical sparse structure, known as a "matching ticket." This process results in improved generalization due to the creation of flat minima and maintains the important linear connectivity between successive solutions, a key strength of the traditional Iterative Magnitude Pruning (IMP) method. SWAMP has demonstrated its ability to outperform other pruning baselines across various datasets and network structures. The technique's success invites further theoretical investigation into why the convex hull of the weight space of these averaged models forms a beneficial low-loss subspace, which could provide deeper insights into the algorithm's effectiveness.

**Strengths:**

- The motivation behind SWAMP is firmly rooted in robust theoretical frameworks, notably the lottery ticket hypothesis and the concept of linear mode connectivity.
- The visualization of the loss landscape in Figure 2 provides a clear illustration of the methodology and supports the validation of the claims made.
- It is clear from the evidence presented in Table 4 that SWAMP is adept at identifying more effective pruning masks.
- Table 2 and 3 demonstrate that SWAMP achieves superior classification accuracy for a designated target sparsity level.

**Weaknesses:**

- The study's reliance on demonstrating the process primarily through wide networks such as WRN and VGG-19, which are not the most parameter-efficient architectures, raises questions about the choice of models. An explanation of why these particular, potentially less efficient, models were selected for this research is needed.

- The improvement in accuracy provided by SWAMP over IMP is modest, as shown in Tables 2 and 3, and this increment is even less pronounced for the ResNet model as evidenced in Table 3. This calls for a discussion on the significance of the marginal gains achieved by SWAMP, particularly when benchmarked against other models.

- The feasibility of achieving an optimal sparse structure with SWAMP, especially for pre-trained models which are commonplace, may entail significant computational costs. It is imperative that the authors address the computational overhead, both in terms of space and time complexity, and the practical constraints when applied to large models, including Transformers. A comprehensive discussion on the limitations is warranted, given that IMP—the foundation of SWAMP—may have its own constraints with larger models.

- The applicability of the proposed method to architectures like Transformers needs clarification. In Table 8, the RoBERTa model exhibits a noticeable performance drop even with less than 50% sparsity. The question arises as to whether this decline is attributed to the inherent limitations of IMP, on which SWAMP is based, or if it pertains to the broader challenges of applying pruning techniques to RoBERTa. Additionally, it would be beneficial to understand whether the principles behind SWAMP remain valid for other models, such as GPT-like architectures, and how they compare with alternative pruning strategies for these models.

**Questions:**

Please refer to Weakness comments

---

> ### Author Response · Authors · 2023-11-14
>
> __[W1] *(The choice of model architectures):*__ We appreciate your attentive review. Throughout the experiments, we aim to verify the effectiveness of SWAMP via various architectures including ResNet (Tiny-ImageNet, ImageNet), Wide-ResNet (Cifar-10, Cifar-100), VGG (Cifar-10, Cifar-100), RoBERTa (GLUE). As per your great suggestion, in Tables R.3 and R.4, we additionally provide experimental results of ResNet-20 on Cifar10 and ResNet-18 on Cifar100, which is a frequently employed experimental setup in prior works [Paul et al., 2023; Yin et al., 2023]. We can readily find that SWAMP works well on not only wide-networks but also on parameter-efficient architectures which highlights that overparameterization is not a crucial factor for the success of SWAMP.
>
> __[W2] *(Marginal gain in Tables 2 and 3):*__ Please refer to General Response B.
>
> __[W3] *(Re. heavy computational costs):*__ Please refer to General Response A.
>
> __[W4] *(Re. transformer-based architectures):*__ Thank you for highlighting the observed performance drop of RoBERTa models beyond 50% sparsity. As shown in Table 8, while both IMP and SWAMP struggle to recover the matching performance, SWAMP still surpasses IMP in terms of accuracy throughout all sparsity levels. In other words, such performance drop stems from inherent limitations of IMP, not a drawback of SWAMP. Moreover, [Chen et al., 2020] supports our findings by empirically confirming that BERT networks suffer to find winning tickets compared to vision task models (See Table 3 in [Chen et al., 2020]). Also, we certainly agree with the reviewer that exploring IMP on GPT-like architectures would be a valuable addition to the community. However, as the focus of our work lie in exploring sparse weight averaging techniques along with empirical verification of the theoretical framework of IMP, we leave it as a future research.
>
> \
> __Table R.3.__ Additional results on Cifar-10 with ResNet-20. Reported values are averaged over three random seeds.
> |     | Sparsity 50% | Sparsity 75% | Sparsity 90% | Sparsity 95% |
> | :-  | :-:          | :-:          | :-:          | :-:          |
> | IMP | 91.57 ± 0.08 | 91.51 ± 0.08 | 90.44 ± 0.16 | 89.07 ± 0.33 |
> | SWAMP | 92.68 ± 0.04 | 92.59 ± 0.20 | 92.03 ± 0.05 | 90.62 ± 0.06 |
>
> __Table R.4.__ Additional results on Cifar-100 with ResNet-18. Reported values are averaged over three random seeds.
> |     | Sparsity 50% | Sparsity 75% | Sparsity 90% | Sparsity 95% |
> | :-  | :-:          | :-:          | :-:          | :-:          |
> | IMP | 75.24 ± 0.51 | 75.38 ± 0.33 | 75.15 ± 0.35 | 74.51 ± 0.45 |
> | SWAMP | 76.25 ± 0.39 | 76.52 ± 0.08 | 76.70 ± 0.03 | 76.43 ± 0.10 |
>
> \
> __References__
> - Chen et al., “The Lottery Ticket Hypothesis for Pre-trained BERT Networks”, NeurIPS 2020.
> - Paul et al., “Unmasking the Lottery Ticket Hypothesis: What's Encoded in a Winning Ticket's Mask?” ICLR 2023.
> - Yin et al., “Lottery Pools: Winning More by Interpolating Ticket without Increasing Training or Inference Cost.” AAAI 2023.

---

> > ### Comment · Reviewer_MGna · 2023-11-21
> > **Response from Reviewer MGna**
> >
> > Thank you for thorough comments and the additional experimental results. However, this reviewer has a couple of lingering questions:
> > 1. Given that unstructured pruning hasn't been widely implemented due to the current limitations in hardware support, I'm curious about the practical benefits of Iterative Magnitude Pruning (IMP) in today's inference context. Why focus on IMP rather than exploring hardware-friendly pruning techniques, which might be more immediately applicable?
> > 2. In what scenarios do you anticipate a significant divergence between the performances of IMP and SWAP? Particularly, I'm interested in knowing if this gap becomes more pronounced with larger models, such as Transformers with over 1 billion parameters. This aspect is crucial, as demonstrating the effectiveness of your proposed technique on such large-scale models, including vision transformers or large language models (LLMs), would significantly elevate the impact of your research. Unfortunately, I couldn't find convincing evidence in the current submission that supports the utility of your technique for these large models.
> >
> > Nevertheless, I acknowledge that your proposed method seems to outperform IMP, as evidenced by the empirical results and the supporting theories you've presented. In light of this, I have decided to increase my score from 5 to 6.

---

> > > ### Author Response · Authors · 2023-11-21
> > >
> > > We appreciate your acknowledgement of the contribution of our work. Please find our additional response to your questions below.
> > >
> > > __[Q1] *(Extension to hardware-friendly pruning methods):*__
> > > Thank you for highlighting (1) the rationale behind choosing IMP and (2) the potential of extending our work to hardware-friendly pruning methods. To address why we specifically chose IMP, as the reviewer pointed out, our focus was to validate whether SWAMP upholds the theoretical characteristics presented in [Paul et al., 2023]; this is crucial in supporting the effectiveness of SWAMP as discussed in Section 3.3. Moreover, in Table 9, we have demonstrated SWAMP's applicability to RigL, one of dynamic pruning methods. Therefore, dynamic pruning methods, which are practically favored upon IMP, may also benefit from SWAMP. Acknowledging the reviewer's emphasis on the applicability of SWAMP, we plan to move this discussion from the appendix to the main section in the final manuscript. While we have not yet explored the combination of structured pruning methods with SWAMP, we will investigate this integration and potentially include it in our final manuscript, as time permits.
> > >
> > > __[Q2] *(Extension to large models over 1B parameters):*__
> > > We certainly agree with your point that the extension to LLM would significantly improve our work. Although we have not yet tested with large models over 1B parameters, we observed that SWAMP enhances IMP in both large-scale ImageNet and GLUE datasets. Moreover, few recent findings suggest that weight averaging is indeed beneficial in training LLMs [Lu et al., 2022; Sanyal et al., 2023]. However, we must acknowledge that this does not automatically imply SWAMP’s compatibility with LLMs. Unfortunately, due to lack of computational resources, we leave it as our future work.
> > >
> > > \
> > > __References__
> > > - Lu et al., “Improving Generalization Pre-trained Language Models via Stochastic Weight Averaging”, EMNLP 2022.
> > > - Sanyal et al., “Understanding the Effectiveness of Early Weight Averaging forTraining Large Language Models”, arXiv preprint 2023.

---

> ### Author Response · Authors · 2023-11-21
> **Reminder**
>
> Dear reviewer-MGna,
>
> We sincerely appreciate your time and the constructive feedback. With the Author/Reviewer discussion deadline approaching, we would be grateful if you let us know whether our responses have addressed your concerns. This will greatly assist us in enhancing our work. For any further clarification, we are more than happy to answer.
>
> Warm regards, \
> Authors

---

### Author Response · Authors · 2023-11-14

__General Response__

We thank all reviewers for their  valuable and constructive comments. They acknowledged that the paper is clearly-written with a good organization (R-tk5N, R-YhLg, R-1f6B, R-Eb3S), and well-motivated (R-MGna, R-tk5N). They also found the paper presents solid arguments along with insightful analysis (R-MGna, R-tk5N, R-YhLg), sufficient ablation studies and experiments (R-tk5N, R-YhLg, R-1f6B, R-Eb3S).

In addition to individual responses, we aim to address the main concerns in the following general response. All of these supplementary experimental outcomes and discussions will be incorporated into the final manuscript.

\
__A. Training cost of the algorithm.__

Reviewers pointed out that the increased training cost of the proposed algorithm could potentially restrict its practicality. Nevertheless, it is worth highlighting that the proposed SWAMP algorithm incurs no additional costs for inference. We believe that the inference cost holds greater significance in real-world scenarios than the training cost - training may take place in an environment with abundant resources and sufficient computation time whereas  deployment frequently requires inference under limited resources such as mobile devices.

Moreover, we further discussed the training costs of the proposed SWAMP algorithm in Section 4.3. To summarize, the extra training costs incurred by (1) multi-particles and (2) iterative procedures in the SWAMP algorithm are effectively alleviated by our recommended strategies, namely parallelization and the SWAMP+ approach. In distributed training environments typically used for large models and datasets, parallelizing multi-particles, as exemplified in our ImageNet experiments, incurs no additional training costs. In addition, SWAMP+ propose to employ multiple particles only in the high-sparsity regime, effectively reducing the substantial training cost mainly attributed to low-sparsity regimes. As shown in Table 6, it is evident that SWAMP+ displays a superior tradeoff between accuracy and training FLOPs compared to IMP and vanilla SWAMP algorithms.

\
__B. Marginal improvement of the algorithm__

We acknowledge that at first glance, the enhancement brought by SWAMP may seem modest. However, we would like to emphasize that SWAMP is simple yet beats both vanilla IMP and the baselines by a noticeable margin. Since IMP is already a SOTA algorithm in the high sparsity regime, we compare two IMP-enhanced baselines in Table R.1 where we display the results of Table 2 in terms of increased accuracy measured in percentage points.

First to highlight is that the enhancement of IMP by SWAMP significantly outperforms the improvement achieved with other baselines. Notably, this gap widens with increasing sparsity levels, which is a remarkable outcome, considering that boosting performance at higher sparsity levels is more challenging. The tendency of SWAMP being more effective in sparser networks is not limited to image classification tasks; it is also observable in our NLP experiments (see Table 8). Additionally, referring to Appendix C.6, we highlight that SWAMP consistently exceeds all baselines in terms of negative log-likelihoods (NLLs), indicating its superior performance in aspects of uncertainty quantification.

Additionally, we provide supplementary outcomes concerning large-scale ImageNet datasets. These results correspond to a pruning ratio of 0.5 for both IMP and SWAMP, in contrast to the 0.8 ratio utilized in Table 3 of the paper. To emphasize the superior generalization capabilities of the SWAMP algorithm, we further provide evaluation results on ImageNet-Sketch and ImageNet-Rendition. We plan to enhance the ImageNet results during the rebuttal period as time permits, and if not possible, we commit to incorporating these results in the final manuscript.

---

> ### Author Response · Authors · 2023-11-14
>
> __Table R.1.__ Improved classification accuracy against vanilla IMP in terms of percentage. The percentage points on the left and right correspond to the results for Cifar-10 and Cifar-100, respectively, as detailed in Table 2.
> | | Sparsity 50% | Sparsity 75% | Sparsity 90% | Sparsity 95% |
> | :-  | :-:          | :-:          | :-:          | :-:          |
> | IMP+SAM       | 0.74 / 0.31 | 0.19 / 0.62 | -0.01 / 0.81 | -1.66 / 0.63 |
> | Lottery Pools | 1.09 / 1.21 | 0.28 / 0.59 |  0.28 / 0.80 | -0.16 / 0.72 |
> | SWAMP         | 1.47 / 2.51 | 0.91 / 2.15 |  0.88 / 2.53 |  0.69 / 2.52 |
>
> \
> __Table R.2.__ Additional results on ImageNet datasets in terms of classification accuracy. Experiments are repeated over three random seeds.
> | Dataset            | Method | 0%    | 45.9% | 68.8% | 80.3% | 86.0% |
> | :-                 | :-     | :-    | :-    | :-    | :-    | :-    |
> | ImageNet-V2        | IMP    | $64.18\pm0.10$ | $64.11\pm0.12$ | $63.78\pm0.03$ | $62.77\pm0.19$ | $61.20\pm0.19$ |
> |                    | SWAMP  |       | $64.34\pm0.38$ | $64.06\pm0.40$ | $63.43\pm0.24$ | $ 61.82\pm0.15$ | $59.44\pm0.07$ |
> | ImageNet-Rendition | IMP    | $35.38\pm0.27$ | $35.04\pm0.25$ | $34.71\pm0.19$ | $34.05\pm0.26$ | $32.85\pm0.29$ |
> |                    | SWAMP  |       | $37.12\pm0.11$ | $36.61\pm0.16$ | $35.61\pm0.22$ | $ 34.14\pm0.31$  |
> | ImageNet-Sketch    | IMP    | $23.90\pm0.02$ | $23.86\pm0.23$ | $23.74\pm0.32$ | $22.88\pm0.26$ | $21.60\pm0.24$ |
> |                    | SWAMP  |       | $25.29\pm0.18$ | $24.95\pm0.17$ | $24.05\pm0.33$ | $22.43\pm0.02$ |

---

### Meta-Review · Area_Chair_M12D · 2023-12-18

**Metareview:**

This paper proposes a nice extension of standard unstructured, iterative neural network pruning algorithms. Specifically, it involves training multiple copies of a network on any given iteration of pruning. It exploits the fact that these networks will remain in the same convex region and prunes the average of the model weights. This appears to give better results than pruning any single copy of the network.

This paper is a bit of a throwback to an earlier and simpler time where we were satisfied to look at various resnets and convnets trained on CIFAR-10. It's a nice improvement to the range of methods that have been studied in those settings, including IMP, RigL, and many others. Although the field is now obsessed with LLMs, I don't think it's a problem that this paper continues to improve on that body of work.

In particular, I'm pleased that someone was able to exploit linear mode connectivity to develop an improved unstructured pruning algorithm; it's something I've been hoping to see.

The method is ridiculously computationally expensive, requiring iterative pruning with multiple copies of the network. But iterative pruning is also ridiculously computationally expensive, and it has always been focused on (a) showing how small networks can get and/or (b) assuming inference will be far more expensive than training. Either way, the cost of finding these sparse networks isn't a huge issue in this literature so long as it's computationally feasible enough to run the appropriate experiments.

The reviewers seem to broadly agree with those sentiments and are in favor of acceptance. I am too.

**Justification For Why Not Higher Score:**

The problem is no longer that interesting or important in the field. This is an elegant and effective enough extension of prior work that I'm still in favor of acceptance, but I personally have a high bar for whether a contribution is worthwhile in this literature. I'd have a hard time believing any paper on unstructured pruning (especially on CIFAR-10) can be worthy of anything more than a poster.

**Justification For Why Not Lower Score:**

The paper is a really nice contribution and the reviewers were all broadly in favor of acceptance.

---

### Decision · Program_Chairs · 2024-01-16

Accept (poster)